# Age-dependent H3K9 trimethylation by dSetdb1 impairs mitochondrial UPR leading to degeneration of olfactory neurons and loss of olfactory function in *Drosophila*

**Francisco Muñoz-Carvajal**[1,2,3], **Nicole Sanhueza**[1], **Mario Sanhueza**[1,4]*, **Felipe A Court**[2,3,5,6]*

[1]Center for Integrative Biology, Faculty of Sciences, Universidad Mayor, Santiago, Chile; [2]Center for Aging Research and Healthy Longevity, Faculty of Sciences, Universidad Mayor, Santiago, Chile; [3]Geroscience Center for Brain Health and Metabolism (GERO), Santiago, Chile; [4]Center for Resilience, Adaptation and Mitigation, Universidad Mayor, Temuco, Chile; [5]Buck Institute for Research on Aging, Novato, United States; [6]Centro Científico y Tecnológico de Excelencia Ciencia and Vida, Fundación Ciencia and Vida, Santiago, Chile

*For correspondence:
mario.sanhueza@umayor.cl (MS);
felipe.court@umayor.cl (FAC)

**Competing interest:** The authors declare that no competing interests exist.

**Abstract** Aging is characterized by a decline in essential sensory functions, including olfaction, which is crucial for environmental interaction and survival. This decline is often paralleled by the cellular accumulation of dysfunctional mitochondria, particularly detrimental in post-mitotic cells, such as neurons. Mitochondrial stress triggers the mitochondrial unfolded protein response (UPR$^{MT}$), a pathway that activates mitochondrial chaperones and antioxidant enzymes. Critical to the efficacy of the UPR$^{MT}$ is the cellular chromatin state, influenced by the methylation of lysine 9 on histone 3 (H3K9). While it has been observed that the UPR$^{MT}$ response can diminish with an increase in H3K9 methylation, its direct impact on age-related neurodegenerative processes, especially in the context of olfactory function, has not been clearly established. Using *Drosophila,* we demonstrate that an age-dependent increase in H3K9 trimethylation by the methyltransferase dSetdb1 reduces the activation capacity of the UPR$^{MT}$ in olfactory projection neurons, leading to neurodegeneration and loss of olfactory function. Age-related neuronal degeneration was associated with morphological alterations in mitochondria and an increase in reactive oxygen species levels. Importantly, forced demethylation of H3K9 through knockdown of dSetdb1 in olfactory projection neurons restored the UPR$^{MT}$ activation capacity in aged flies, and suppressed age-related mitochondrial morphological abnormalities. This, in turn, prevented age-associated neuronal degeneration and rescued age-dependent loss of olfactory function. Our findings highlight the effect of age-related epigenetic changes on the response capacity of the UPR$^{MT}$, impacting neuronal integrity and function. Moreover, they suggest a potential therapeutic role for UPR$^{MT}$ regulators in age-related neurodegeneration and loss of olfactory function.

## Editor's evaluation

This important manuscript supports a model that age-dependent increases in H3K9me3, by dSetdb1, repress the mitochondrial UPR, leading to neuronal degeneration and olfactory decline.

The evidence is convincing but a few minor limitations remain, such as a mechanistic link between H3K9me3 and UPRmt repression.

## Introduction

Aging is associated with a time-dependent organ dysfunction that increases the vulnerability of an organism to various forms of stress, ultimately leading to organism death (*Olofsson et al., 2021*; *Kondo et al., 2020*; *Fatuzzo et al., 2023*; *Lucke et al., 2020*; *Hussain et al., 2018*; *Pellegrino et al., 2013*; *Kim and Sieburth, 2018*). Aging induces alterations across various physiological systems, with the olfactory system being particularly affected (*Olofsson et al., 2021*). Olfaction, the sense of smell, is essential for detecting environmental odors crucial for feeding, reproductive and survival behaviors (*Kondo et al., 2020*; *Fatuzzo et al., 2023*). Importantly, dysfunction in olfaction is emerging as one of the early signs of neurodegenerative diseases, including Alzheimer's and Parkinson's disease (*Jenkins et al., 2021*; *Couvillion et al., 2016*; *Nargund et al., 2012*). Research on *Drosophila melanogaster* has shown that the age-related decline in odor response is influenced by the functional state of mitochondria in olfactory projection neurons (OPNs) (*Lucke et al., 2020*). This decline is accompanied by defects in neuronal integrity and a decrease in synaptic proteins (*Hussain et al., 2018*), which correlates with a reduction in mitochondria and an increase in ROS (*Hussain et al., 2018*), but the underlying mechanisms has not been defined.

Under compromised mitochondrial integrity or function, cells engage in a transcriptional response known as the mitochondrial unfolded protein response (UPR^MT) (*Tian et al., 2016*). This mitochondrial program can be activated by the impairment of the electron transport chain (ETC), alteration of mitochondrial dynamics, accumulation of unfolded proteins, reduction of mitochondrial DNA, or reduction of mitochondrial chaperones or protease (*Pellegrino et al., 2013*; *Kim and Sieburth, 2018*; *Jenkins et al., 2021*; *Couvillion et al., 2016*). Upon UPR^MT pathway engagement, the transcription factor ATFS-1 (*C. elegans* ortholog of mammalian ATF5 and *Drosophila* crc) translocates from the mitochondria to the nucleus (*Fiorese et al., 2016*; *Nargund et al., 2012*). In the nucleus, ATFS-1, DVE-1, and UBL-5 interact to reorganize the chromatin structure, enabling activation of the nuclear transcription of mitochondrial chaperones, including hsp-60 and hsp-6, and the protease clpp-1 and lonp. This coordinated transcriptional response restores mitochondrial function under stress conditions by metabolic adaptations and enhancing mitochondrial biogenesis (*Haynes et al., 2007*; *Benedetti et al., 2006*; *Tian et al., 2016*). In mammals, the functional homolog of ATFS-1 is activating transcription factor 5 (ATF5). ATF5, like its *C. elegans* counterpart, contains both nuclear and mitochondrial localization signals, allowing it to shuttle between compartments depending on mitochondrial stress levels (*Fiorese et al., 2016*), and when expressed in worms without ATFS-1, it induced hsp-60 during mitochondrial stress but not during endoplasmic reticulum (ER) stress (*Fiorese et al., 2016*). Upon mitochondrial dysfunction, ATF5 accumulates in the nucleus. and induces the expression of HSP60, mtHSP70, LONP1 (*Fiorese et al., 2016*) and numerous genes involved in mitochondrial biogenesis, metabolism, protein folding, and ROS detoxification (*Nargund et al., 2012*; *Wu et al., 2014*). Notably, the *Drosophila* gene crc, which shares homology with both ATF5 and ATF4, is a key regulator of the UPR^MT. While crc has been primarily characterized in the context of the ER unfolded protein response (UPR^ER), recent evidence suggests a broader role in cellular stress responses, including the UPR^MT. For instance, the mammalian homolog ATF4 has been shown to regulate the UPR^MT [16], suggesting a potential crosstalk between these stress pathways. Furthermore, studies have implicated crc in mitochondrial function and maintenance (*Celardo et al., 2017*; *Hunt et al., 2019*). Although the precise mechanisms by which crc regulates the UPR^MT in *Drosophila* remain to be fully elucidated, its homology to ATF5 and its involvement in mitochondrial processes strongly suggest a conserved role in mitochondrial stress response.

Chromatin remodeling is crucial for UPR^MT regulation, with the epigenetic state of lysine 9 on histone 3 (H3K9) serving as a critical regulator of the response (*Tian et al., 2016*; *Merkwirth et al., 2016*; *Ono et al., 2017*; *Sobue et al., 2017*). Changes in H3K9 methylation by the methyltransferase MET-2, the *C. elegans* ortholog of human SETDB1, modify UPR^MT-related loci exposure, modulating binding of UPR^MT regulators DVE-1 and ATFS-1 (*Sobue et al., 2017*). In addition, enzymes that remove methyl groups from H3K9 significantly influence UPR^MT activation. For example, demethylases JMJD-3.1 and JMJD-1.2 remove trimethylation from H3K9me3 and H3K27me3, enabling UPR^MT activation

(*Merkwirth et al., 2016*; *Sobue et al., 2017*). Recent studies revealed the effects of H3K9me3 methylation on UPR^MT activation and mitochondrial function across species. In *C. elegans,* the epigenetic factors BAZ-2 and SET-6, which regulate H3K9me3 levels, have conserved roles in impacting aging processes through mitochondrial function (*Yuan et al., 2020*). In mice, deletion of Baz2b, a homologue of BAZ-2, shows beneficial effects on mitochondrial function and cognitive abilities, indicating a conserved mechanism across species that influences aging and healthspan through mitochondrial function and epigenetic regulation (*Hunt et al., 2019*). Accordingly, in the hippocampus of aged mice, H3K9me3 levels rise with age, a change associated with age-related cognitive decline (*Snigdha et al., 2016*). Age-related increases in H3K9me3 have also been observed in the brains of aged *Drosophila* and muscle stem cells of aged mice (*Wood et al., 2010*; *Schwörer et al., 2016*). While previous research has linked changes in methylation levels to age-related functional decline, the specific role of these epigenetic alterations in the context of neuronal degeneration through reduced UPR^MT activation capacity remains to be fully elucidated. This study aims to bridge this gap by providing detailed insights into how epigenetic mechanisms, particularly methylation changes, directly contribute to the aging-associated loss of olfactory function and neuronal degeneration by impacting the UPR^MT pathway and mitochondrial function.

Here, we employed behavioral, molecular, and morphological methodologies to investigate whether epigenetic regulation of UPR^MT is linked to neurodegeneration in the aging brain and its involvement in age-associated olfactory decline. To this end, we utilized the OPNs in the adult *Drosophila* antennal lobe (AL), which exhibit age-related neurodegeneration correlating with functional neuronal decline (*Hussain et al., 2018*). Our results demonstrate that with aging, there is a decline in the response capacity of the UPR^MT in OPNs, functionally associated with a dSetdb1-dependent increase in H3K9me3 levels. Genetic inhibition of dSetdb1 reduces H3K9me3 levels, enabling the activation of UPR^MT, restoring mitochondrial oxidation to youthful levels, and preventing age-associated degeneration of OPNs. This effect is particularly evident in the somas located in the AL and the presynaptic connections of the lateral horn (LH) in the *Drosophila* brain. Importantly, maintaining UPR^MT activation during aging preserved olfactory function. These findings underscore the critical role of epigenetic regulation, specifically through dSetdb1 and H3K9me3, in modulating neuronal integrity and sensory function during aging.

## Results

### UPR^MT response capacity decreases with aging in the *Drosophila* antennal lobe

To study the modulation of UPR^MT along aging and its association to olfactory function, we first generated reporters based on the expression of the fluorescent protein dsRed under the promoters of chaperones hsp60 and hsc70-5, which specifically responds to UPR^MT stimuli across species and has been effectively used to indicate UPR^MT activation (*Morrow et al., 2016*; *Owusu-Ansah et al., 2013*; *Borch Jensen et al., 2017*; *Kumar et al., 2022*; *Yoneda et al., 2004*). We focused on the AL, the functional homolog of the vertebrate olfactory bulb where olfactory projection neurons process sensory inputs (*Figure 1A*). For our analysis, we used a standardized 3D surface-based quantification method and confirmed that the Hsp60::dsRed reporter is induced by mitochondrial stress (*Figure 1—figure supplements 2 and 3*). In young flies (0 dpe), a low-intensity signal from the Hsp60::dsRed and Hsc70-5::dsRed reporters was detected under control conditions, which significantly increased upon exposure to the UPR^MT activators paraquat (PQ) and doxycycline (Doxy) (*Figure 1A–D*). Importantly, this increase in Hsp60::dsRed signal was not induced by non-specific mitochondrial stressor tunicamycin, which activates UPR^ER and the ER-stress specific reporter Xbp1::GFP (*Ryoo, 2015*; *Figure 1—figure supplement 3A–D*). To further evaluate the specificity of the UPR^MT reporter, we pan neuronally downregulated the UPR^MT nuclear activators *dve, ubl,* and *crc*. Pan-neuronal downregulation driven by *Elav-Gal4* of these UPR^MT activators significantly reduced the Hsp60::dsRed response to PQ compared to control flies (*Figure 1E and F*), consistent with these factors contributing to UPR^MT-dependent reporter activation. Additionally, knockdown of crc and ubl directly impaired the UPR^MT response, leading to reduced expression of Hsp60::dsRed reporter even under basal conditions (*Figure 1F*). We next used the Hsp60::dsRed reporter to evaluate UPR^MT activity during aging in the *Elav-Gal4* driven CD8::GFP-labeled neurons of *Drosophila* AL. Compared to the robust signal triggered by PQ in young

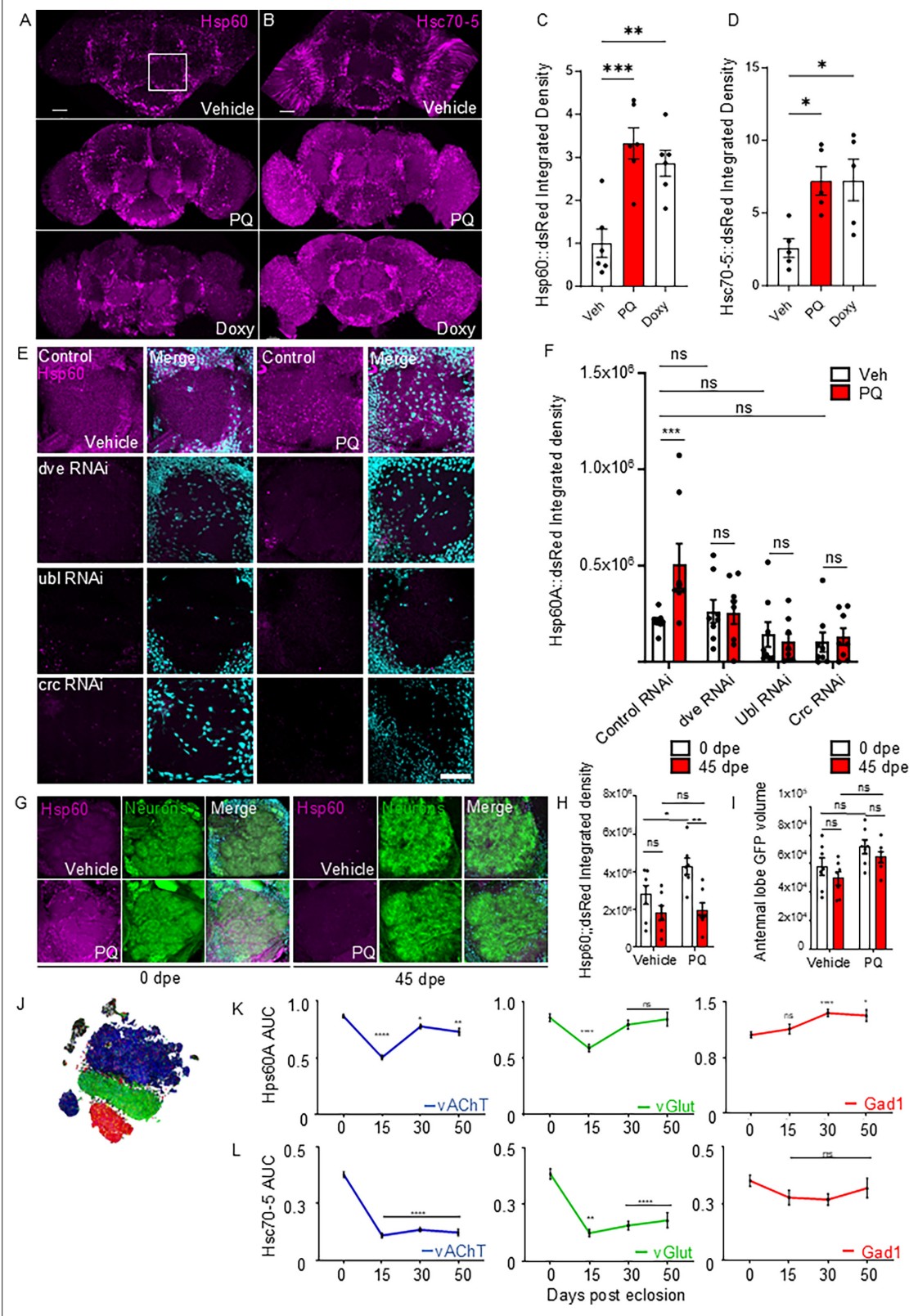

**Figure 1.** UPR^MT-dependent activation of the Hsp60::dsRed reporter in the antennal lobe (AL) of *Drosophila*. (**A**) Representative confocal images of whole *Drosophila* brains flies expressing the UPR^MT reporter Hsp60::dsRed (magenta). Flies were treated for 48 hr with vehicle, paraquat (paraquat PQ, 10 µM), or doxycycline (Doxy, 100 µM). The white box indicates the quantified region of the AL. Scale bar, 20 µm. (**B**) Confocal images of the whole *Drosophila* brains expressing the UPR^MT reporter Hsc70-5::dsRed (magenta) under the same treatment conditions as in (**A**). (**C**) Quantification of

*Figure 1 continued on next page*

*Figure 1 continued*

Hsp60::dsRed integrated density in the AL. (n=6, Veh vs. PQ: $p=0.0003$ and Veh vs. Doxy: $p=0.0024$).(**D**) Quantification of Hsc70-5::dsRed integrated density in the AL. (n=5, Veh vs. PQ: $p=0.0198$ and Veh vs. Doxy: $p=0.0184$). (**E**) Hsp60::dsRed in AL with pan-neuronal RNAi knockdown of dve, ubl, crc, post treatment with PQ or vehicle; magenta shows the Hsp60A-dsRed signal, cyan for DAPI. Scale bar, 20 μm. (**F**) Integrated density of Hsp60::dsRed, Two-way ANOVA Dunnett test between veh vs PQ treated flies: Control ($p=0.0038$, n=8), dve ($p>0.9999$, n=8), crc ($p>0.9999$, n=8), ubl ($p>0.9999$, n=8). (**G**) Representative images Hsp60::dsRed reporter in GFP-labeled neurons of the AL of flies treated with 10 mM PQ or vehicle for 48 hr. Hsp60::dsRed in magenta, Pan-neuronal GFP in green, and DAPI in cyan. Scale bar, 20 μm. (**H**) Integrated density of Hsp60::dsRed in GFP-labeled neurons from 0 (white bars) to 45 dpe (red bars). Two-way ANOVA with Dunnett's multiple comparisons between ages for vehicle ($p=0.2319$, n=7) and for PQ treatment ($p=0.0017$, n=7). For the difference between treatments: vehicle and PQ at 0 dpe ($p=0.0404$, n=7), and at 45 dpe ($p=0.9581$, n=7). (**I**) Neuronal volume (μm$^3$) in AL from 0 (white bars) to 45 dpe (red bars). Two-way ANOVA with Dunnett's multiple comparisons between 0 and 45 dpe flies for vehicle ($p=0.4255$, n=7) and PQ ($p=0.5076$, n=7). Comparing vehicle to PQ treatment showed no significant difference at 0 dpe ($p=0.1133$, n=7) and 45 dpe ($p=0.0857$, n=7). (**J**) Dot plot visualizing vAChT (blue), vGlut (green), and Gad1 (red) for UPR$^{MT}$ activation analysis via single-cell RNA seq data. (**E**) AUC scores of Hsp60 expression: vAChT (0 vs 50 dpe, $p=0.0049$, n=576), vGlut (0 vs 50 dpe, $p=0.9998$, n=168), and Gad1 (0 vs 50 dpe, $p=0.0191$, n=168). (**F**) AUC scores of Hsc70-5 expression: vAChT (0 vs 50 dpe, $p<0.0001$, n=576), vGlut (0 vs 50 dpe, $p<0.0001$, n=168), and Gad1 (0 vs 50 dpe, $p=0.9914$, n=168). For panels C–I, each n represents one brain from an individual animal. For panels J–L, each n represents a single cell. Bars represent mean ± SEM. Statistical significance is denoted as ****$p<0.0001$; ***$p<0.001$; **$p<0.01$; *$p<0.05$; ns>0.05.

The online version of this article includes the following figure supplement(s) for figure 1:

**Figure supplement 1.** Quantification of the fluorescent signal of Hsp60::dsRed reporter using 3D reconstructed surface in the antennal lobe of *Drosophila* brain.

**Figure supplement 2.** The Hsp60A regulatory region::dsRed transcriptional reporter is induced by mitochondrial stress in vivo.

**Figure supplement 3.** The *Hsp60::dsRed* reporter is specifically activated by mitochondrial stress, not by endoplasmic reticulum (ER) stress.

flies, aged flies (45 dpe) did not exhibit Hsp60::dsRed reporter activation in AL neurons after PQ (*Figure 1G–H*). Importantly, no significant changes in GFP-labeled neuronal volume were observed in aged versus young flies or after PQ treatment (*Figure 1I*). This data suggests that the ability to trigger UPR$^{MT}$ activity declines with advanced age.

We then explore the age-dependent endogenous expression of UPR$^{MT}$-associated chaperones Hsp60 and Hsc70-5 using Scope, a single-cell gene expression repository of brain cells from *Drosophila* at different ages (http://scope.aertslab.org) (*Davie et al., 2018*). The *Drosophila* brain consists of three major groups of neurons: glutamatergic, GABAergic, and cholinergic neurons (*Figure 1J*). Single-cell RNA sequencing data of whole fly brains revealed that Hsp60A expression in cholinergic neurons fluctuates with age, while Hsc70-5 expression decreases (*Figure 1K–L*). These age-related changes in chaperone expression were not observed in glutamatergic or GABAergic neurons, suggesting a cell-type-specific vulnerability in UPR$^{MT}$ regulation.

## Epigenetic regulation of the UPR$^{MT}$ by dSetdb1 in the AL of *Drosophila* brain

It has been previously demonstrated that trimethylation of H3K9 increases during *Drosophila* aging (*Wood et al., 2010*), a phenomenon that mirrors observations in other species. To assess whether methylation levels of H3K9 can modulate UPR$^{MT}$ activation in flies, we studied flies with pan-neuronal knockdown of dSetdb1, a specific H3K9 methyltransferase. Our data demonstrates that pan-neuronally downregulating dSetdb1 prevents the age-associated increase in H3K9 trimethylation in homogenates of *Drosophila* heads (*Figure 2A*). We then investigated the role of H3K9 trimethylation in UPR$^{MT}$ activation by examining the Hsp60::dsRed reporter in flies with a ubiquitous loss of function of dSetdb1 (egg$^{235}$), which harbors a point mutation at the donor splice site of intron 4, leading to a premature stop codon and a truncated, non-functional protein. Control flies exhibited similar basal levels of Hsp60::dsRed signal in the AL of young and aged flies. Similarly, we observed no differences in the reporter signal for young flies in which dSetdb1 was downregulated (*Figure 2B*), consistent with the low levels of H3K9 trimethylation observed in young animals (*Figure 2A*). However, the Hsp60::dsRed signal in aged dSetdb1 mutants was significantly higher compared to age-matched control flies (*Figure 2B–C*). These results were further confirmed using the Hsc70-5::dsRed UPR$^{MT}$ reporter (*Figure 2D–E*). These data suggest that dSetdb1 contributes to age-dependent H3K9 trimethylation, and its reduced function in the AL of aged flies correlates with a basal increase in UPR$^{MT}$ activity.

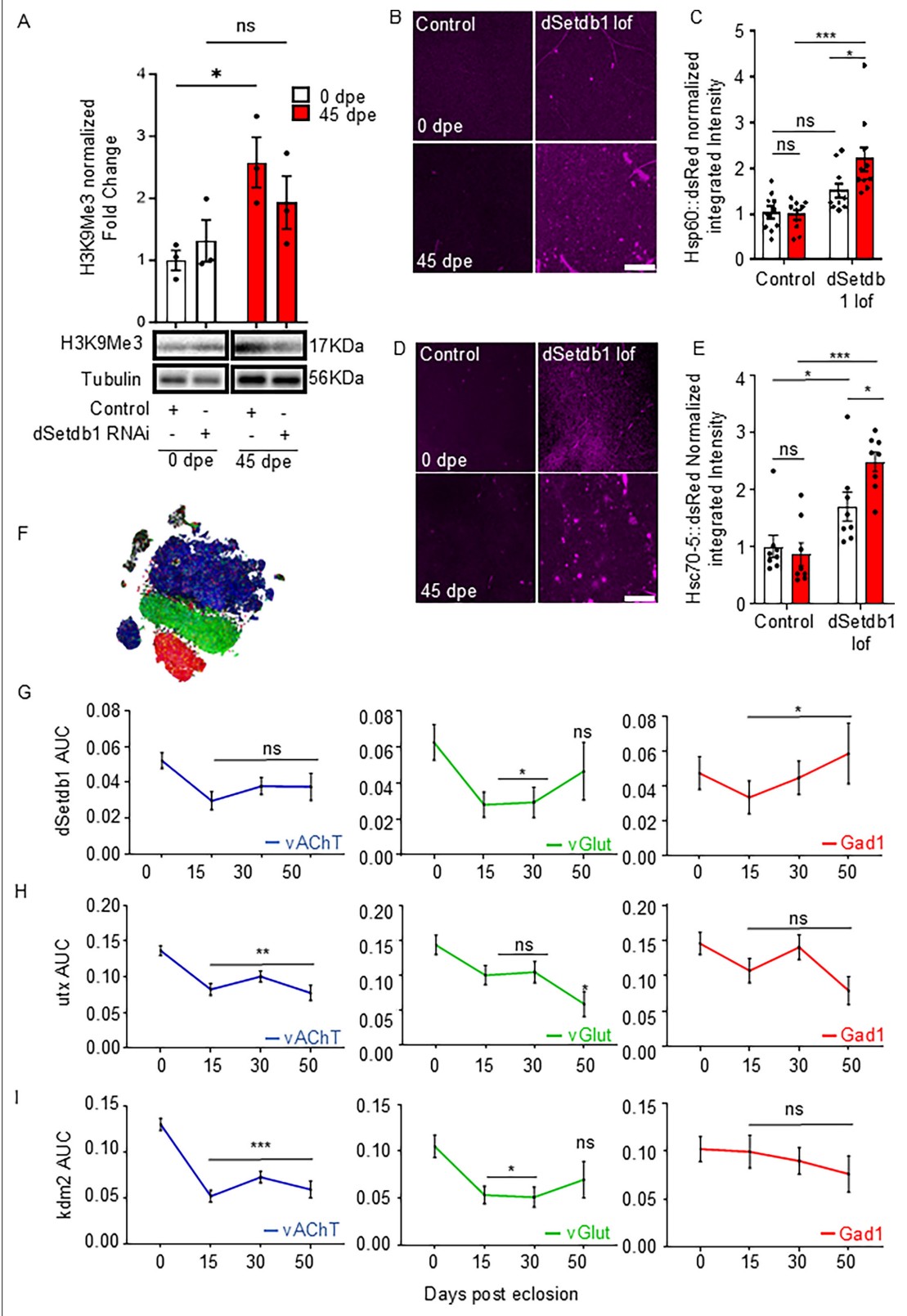

**Figure 2.** dSetdb1 negatively regulates UPR^MT in aging through increasing H3K9me3 levels in antennal lobe (AL) of *Drosophila*. (**A**) Western blot for H3K9me3 levels with pan-neuronal downregulation of dSetdb1. Two-way ANOVA Bonferroni's multiple comparisons test results: Control at 0 vs 45 days post eclosion (dpe) ($p$=0.0255, n=3), dSetdb1 RNAi at 0 vs 45 dpe ($p$=0.4951, n=3), Control vs. dSetdb1 RNAi at 0 dpe ($p$>0.9999, n=3), and at 45 dpe ($p$=0.4556, n=3). n=20 fly heads. (**B**) Representative confocal images of AL from UPR^MT reporter flies with dSetdb1 loss of function (lof), displaying

*Figure 2 continued on next page*

*Figure 2 continued*

Hsp60A::dsRed in magenta. Scale bar, 20 µm. (**C**) Normalized integrated density of Hsp60A::dsRed. Two-way ANOVA Bonferroni's multiple comparisons test between 0 vs 45 dpe within *dSetb1 l of* genotype ($p<0.0001$, n=10) and control ($p>0.9999$, n=10). (**D**) Confocal images of Hsc70-5::dsRed in AL from flies at 0 and 45 dpe with dSetdb1 loss of function. Scale bar, 20 µm. (**E**) Integrated density of Hsc70-5::dsRed normalized to control values. Two-way ANOVA Bonferroni's multiple comparisons between 0 vs 45 dpe within *dSetdb1 Lof* genotype ($p<0.0001$, n=8) and control ($p=0.8633$, n=8). (**F**) Dot plot from Scope single-cell RNA-seq analysis depicting neuronal types: vAChT (blue), vGlut (green), and Gad1 (red). (**G**) AUCell scores for dSetdb1 expression in single neurons at 0, 15, 30, and 50 dpe. One-way ANOVA with Dunnett's multiple comparison against 0 dpe. vAChT neurons: 50 dpe ($p=0.2906$, n1=2932, n2=656). vGlut neurons: 50 dpe ($p=0.7386$, n1=712, n2=172). Gad1 neurons: 50 dpe ($p=0.8916$, n1=576, n2=197). (**H**) AUCell scores for utx expression in single neurons at 0, 15, 30, and 50 dpe. One-way ANOVA with Dunnett's multiple comparison against 0 dpe. vAChT neurons: 50 dpe ($p=0.0001$, n1=2932,, n2=656). vGlut neurons: 50 dpe ($p=0.321$, n1=712, n2=172). Gad1 neurons: 50 dpe ($p=0.0688$, n1=576, n2=197). (**I**) AUCell scores for kdm2 expression in single neurons at 0, 15, 30, and 50 dpe. One-way ANOVA with Dunnett's multiple comparison against 0 dpe. vAChT neurons: 50 dpe ($p<0.0001$, n1=2932, n2=656). For vGlut neurons: 50 dpe ($p=0.321$, n1=712, n2=172). For Gad1 neurons: 50 dpe ($p=0.667$, n1=576, n2=197). For panels C and E, each n represents one brain from an individual animal. For panels G–I, n1 and n2 represent single cells from each age group. All error bars represent mean ± SEM. P-value: ****$p<0.0001$; ***$p<0.001$; **$p<0.01$, *$p<0.05$ and ns>0.05.

The online version of this article includes the following source data for figure 2:

**Source data 1.** PDF file containing original western blots for *Figure 2A*, indicating the relevant bands and treatments.

**Source data 2.** Original files for western blot analysis displayed in *Figure 2A*.

To understand the relevance of H3K9me3-related genes in a neuron-specific context, we then analyzed single-cell data from Scope to assess the expression levels of dSetdb1, as well as the H3K9 demethylases Utx and Kdm2 (*Merkwirth et al., 2016*; *Herz et al., 2010*). In vAChT neurons, dSetdb1 expression remains constant throughout aging (*Figure 2G*). However, both Utx and Kdm2 exhibit an age-dependent decrease in expression (*Figure 2H and I*). Together, this data suggests that the age-dependent reduction in H3K9 demethylation enzymes could be associated with higher levels of H3K9me3 in the aged *Drosophila* brain, which in turn might contribute to the age-related decrease in UPR^MT activity.

## Age-dependent decline in olfactory function depends on the epigenetic modulation of the UPR^MT

Having established that the decline in UPR^MT activity in aged *Drosophila* is linked to elevated levels of H3K9me3, we then explored the potential link between UPR^MT activation and olfactory function in *Drosophila*. The ability to discern between odors diminishes with age in flies, a quantifiable phenomenon through the olfactory T-maze (*Figure 3A–B*). Therefore, we genetically downregulated the UPR^MT transcriptional activators dve, ubl, or crc and studied the ability of flies to discriminate odors throughout their lifespan. Remarkably, young flies with the knockdown of the nuclear activators of the UPR^MT exhibited a reduced olfactory capacity to discriminate an abrasive odor compared to control animals (*Figure 3C*). Aged flies with the knockdown of dve, ubl, or crc did not show significantly different olfactory function compared to age-matched controls or to young flies from the same genotype (*Figure 3C*). These data support the involvement of UPR^MT transcriptional activators in olfactory discrimination, consistent with prior work. We next explored the impact of the epigenetic regulation of UPR^MT in neuronal functionality. To this end, we generated flies with pan-neuronal knockdowns of the H3K9 methyltransferase dSetdb1 and demethylases Kdm2 or Utx. Consistent with our previous observations, young flies with downregulated dSetdb1 did not show a difference in H3K9me3 levels compared to controls. In contrast, downregulation of demethylases Utx or Kdm2 led to increased H3K9me3 levels, highlighting their distinct regulatory roles (*Figure 3D*). To determine if H3K9me3 levels alter neuronal functionality in the olfactory system, we assessed olfactory function in flies with downregulation of dSetdb1, Utx, or Kdm2. In young flies, pan-neuronal knockdown of dSetdb1 showed no significant difference from control flies. However, aged dSetdb1 mutant flies exhibited improved olfactory function, with no significant difference when compared with young flies (*Figure 3E*). On the other hand, pan-neuronal knockdown of Utx or Kdm2 impaired olfactory function in young flies, with an odor discrimination capacity similar to aged control flies (*Figure 3E*), indicating that age-dependent increases in H3K9me3 progressively affect olfactory function. These phenotypes are not caused by a shortened lifespan, as the survival curves for kdm2 and utx are not significantly different from those of the controls. (*Figure 3—figure supplement 1*).

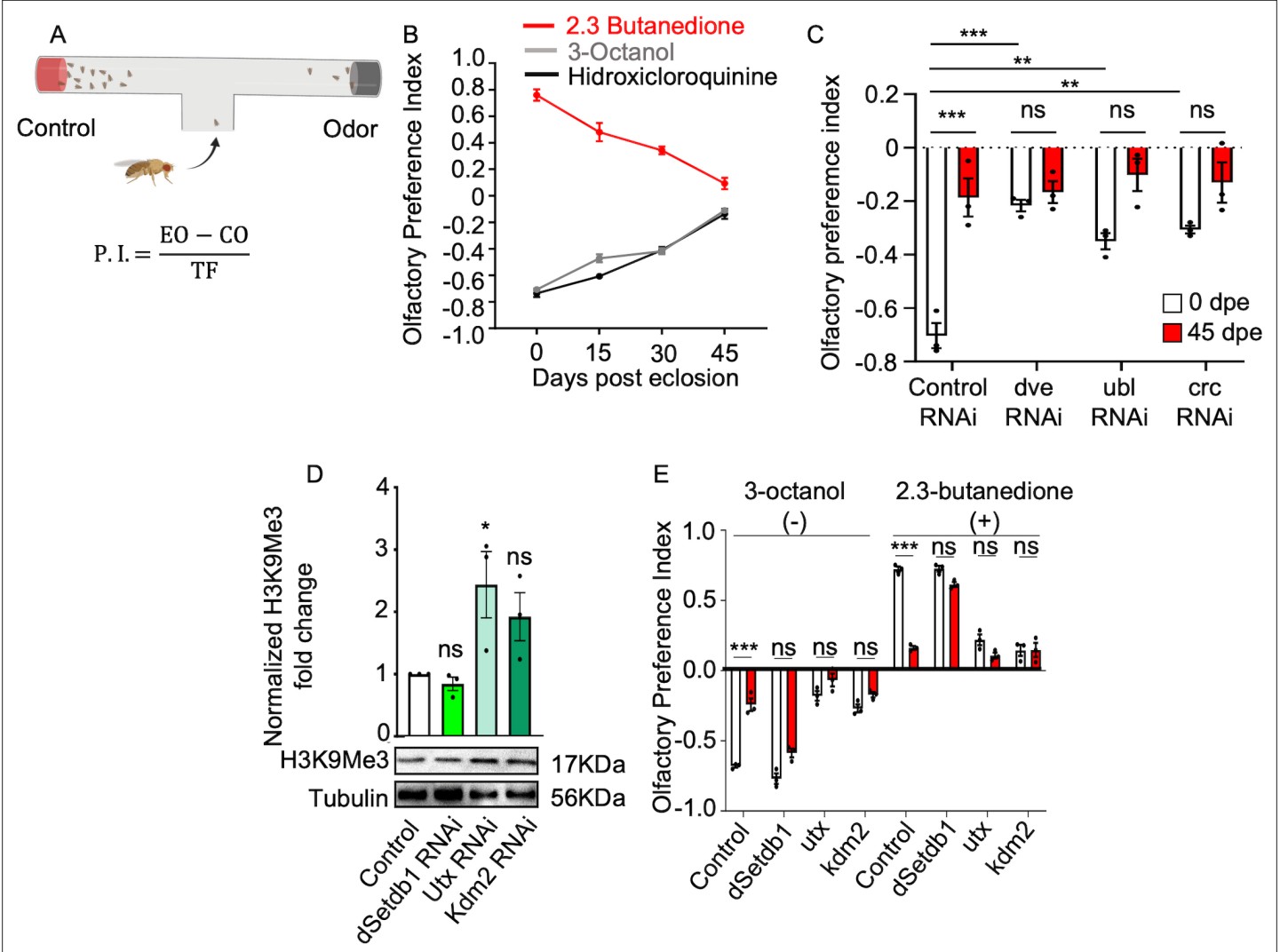

**Figure 3.** dSetdb1 pan-neuronal downregulation preserves olfactory function in aging. (**A**) Olfactory T-maze was used to perform the olfactory behavioral test. Flies are presented to an experimental odor or vehicle. Flies have 60 s to discriminate between odors and go to an arm of the T-maze. At the end of the time, an image is acquired, and the preference index is calculated for every trial; every dot corresponds to 10 trials of 15 flies each. (**B**) Olfactory preference index shows the aging-associated functional decline in the olfactory system. (**C**) Olfactory preference index in flies with pan-neuronal downregulation of dve, ubl, and crc. n=3 populations of 10 flies each. (**D**) Western blot analysis of H3K9me3 levels in young flies with pan-neuronal downregulation of dSetdb1 (green), utx (gray), and kdm2 (calypso). Each n represents homogenized pools of 20 fly heads. One-way ANOVA Bonferroni's multiple comparisons test results: Control vs. dSetdb1 RNAi ($p=0.9738$, n=3), Control vs. utx RNAi ($p=0.0391$, n=3), Control vs. kdm2 RNAi ($p=0.1951$, n=3). (**E**) Olfactory preference indices for 0 and 45 days post eclosion (dpe) flies with downregulation of dSetdb1, utx, and kdm2, exposed to odors 3-octanol (-) and 2,3-butanedione (+). Two-way ANOVA Bonferroni's test results for 3-octanol (-), control at 0 vs 45 dpe ($p<0.0001$, n=3), dSetdb1 ($p=0.0624$, n=3), utx ($p=0.1356$, n=3), kdm2 ($p=0.298$, n=3); for 2,3-butanedione (+), control ($p<0.0001$, n=3), dSetdb1 ($p=0.1356$, n=3), utx ($p=0.1374$, n=3), kdm2 ($p>0.9999$, n=3). All error bars represent mean ± SEM.

The online version of this article includes the following source data and figure supplement(s) for figure 3:

**Source data 1.** PDF file containing original western blots for *Figure 3D*, indicating the relevant bands and treatments.

**Source data 2.** Original files for western blot analysis displayed in *Figure 3D*.

**Figure supplement 1.** dSetdb1 pan-neuronal downregulation does not increase max survival, but increases healthspan (**A**) Survival curve of flies bearing pan-neuronal downregulation of dSetdb1 (green), utx (gray), and kdm2 (calypso) compared to control (black).

While the pan-neuronal knockdown of dSetdb1 improved olfactory behavior, this could be an indirect consequence of a general improvement in organismal health (*Figure 3—figure supplement 1*). Similarly, dve, ubl, and crc were tested using single RNAi reagents (*Figures 1E-F, 3C and 4L*); therefore, conclusions regarding these factors remain limited and await further independent validation, and

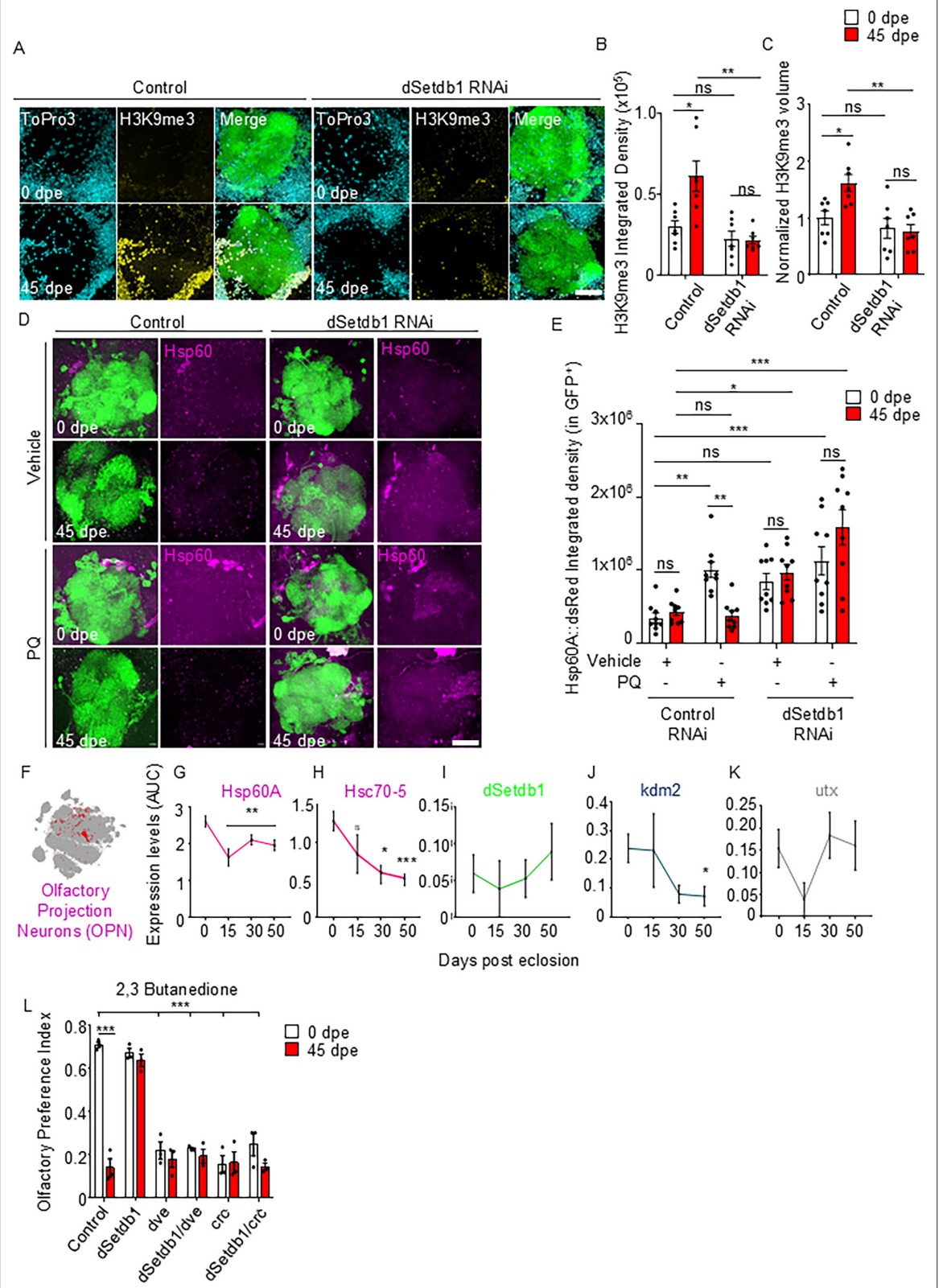

**Figure 4.** dSetdb1 downregulation preserves olfactory projection neurons (OPNs) function in aging by reducing H3K9me3 and enabling UPR^MT.
(**A**) Representative images of GFP-tagged OPNs in the antennal lobe (AL) of 0 and 45 days post eclosion (dpe) flies bearing the downregulation of dSetdb1 under the control of the GH146 driver. Nuclei are in cyan (ToPro3), H3K9me3 is in yellow, and the right panel is a merge of three channels with OPNs in green. Scale bar, 20 μm. (**B**) Analysis of H3K9me3 integrated density in the nuclei of GFP-tagged OPNs across aging in *Drosophila* with dSetdb1

*Figure 4 continued on next page*

Figure 4 continued

knockdown. Using two-way ANOVA with Bonferroni's correction, control flies showed a significant reduction in density from 0 to 45 dpe ($p$=0.0128, n=7), a difference lost at dSetdb1 RNAi flies ($p$=0.5022, n=7). (**C**) Quantification of H3K9me3 volume, specifically in nuclei of GFP signal, normalized to control of 0 dpe untreated flies. White bars represent 0 dpe flies and red bars 45 dpe flies, n=7. Two-way ANOVA Bonferroni's multiple comparisons showed for control flies of 0 vs 45 dpe $p$=0.0057; or dSetdb1 RNAi group of 0 vs 45 dpe ($p$>0.9999), Control vs dSetdb1 RNAi at 0 dpe $p$=0.0604, for 45 dpe, Control vs dSetdb1 RNAi $p$=0.0004 (**D**) Representative images of OPNs labeled with GFP (green) bearing the UPR$^{MT}$ reporter Hsp60::dsRed (magenta) in the AL of 0 and 45 dpe flies treated with paraquat (PQ) or vehicle for 48 hr. Scale Bar, 20 µm. (**E**) Quantification of Integrated density of Hsp60A::dsRed specifically in the GFP-labeled neurons in the AL of 0 and 45 dpe flies treated with vehicle or PQ 10 mM for 48 hr. Two-way ANOVA Bonferroni's test result at 0 dpe between Control Vehicle vs. Control PQ $p$=0.0027, n=9; Control Vehicle vs. dSetdb1 Veh $p$=0.0291, n=9; Control Vehicle vs. dSetdb1 PQ $p$=0.0003, n=9 and at 45 dpe between Control Vehicle vs. Control PQ $p$=0.9872, n=9; Control Vehicle vs. dSetdb1 Veh $p$=0.0154, n=9; Control Vehicle vs. dSetdb1 PQ $p$<0.0001, n=9. (**F**) Dot plot of expression cluster showing specifically OPN cluster in red. One-way ANOVA with Dunnett's multiple comparisons test against 0 dpe was performed for (**G**) Hsp60A Expression: At 0 vs 50 dpe $p$=0.0102, n1=60, n2=38, and (**I**) Hsc70-5 expression 0 vs 50 dpe $p$<0.0001, n1=60, n2=38. (**I–K**) Expression levels of dSetdb1, kdm2, and utx in OPNs through aging. One-way ANOVA with Dunnett's multiple comparisons test against 0 dpe was performed, (**I**): dSetdb1 expression 0 vs 50 dpe ($p$=0.0102, n1=84, n2=56). (**J**) utx expression 0 vs 50 dpe ($p$=0.9996, n1=84, n2=56). (**K**) kdm2 0 vs 50 dpe ($p$=0.0425, n1=84, n2=56). Each n represents the AUC values for expression in a single cell. (**L**) Olfactory preference index of flies bearing the GH146 Gal4 driven knockdown of dSetdb1, dve, crc, and the double knockdown of dSetdb1/dve and dSetdb1/crc, respectively. White bars are 0 dpe flies and red bars are 45 dpe flies. n=3. Results from a Two-way ANOVA Bonferroni's multiple comparisons test are as follows: Control at 0 vs 45 dpe: $p$<0.0001, n=3, dSetdb1 at 0 vs 45 dpe: $p$>0.9999, n=3; dve at 0 vs 45 dpe: $p$>0.9999, n=3; dSetdb1/dve at 0 vs 45 dpe: $p$>0.9999, n=3; crc at 0 vs 45 dpe: $p$>0.9999, n=3; dsetdb1/crc at 0 vs 45 dpe: $p$=0.2606, n=3. P-value: ****$p$<0.0001; ***$p$<0.001; **$p$<0.01, *$p$<0.05 and ns >0.05. For panels B, C, and E, each n represents one brain from an individual animal. For panels G–K, n1 and n2 represent single cells from each age group. In panel L, each n represents one population of 10 flies. All error bars represent mean ± SEM.

The online version of this article includes the following figure supplement(s) for figure 4:

**Figure supplement 1.** Validation of the *dSetdb1* function in aged flies using independent genetic tools.

**Figure supplement 2.** Olfactory projection neuron (OPN)-specific dSetdb1 knockdown restores age-dependent UPR$^{MT}$ activation.

potential off-target effects cannot be fully excluded. Therefore, to distinguish indirect organism-wide effects from a direct impact on olfactory circuitry and to robustly validate our primary finding, we next targeted dSetdb1 function specifically within OPNs using the GH146-Gal4 driver (***Figure 4—figure supplement 1***). To robustly validate that reducing the dSetdb1 function underlies the preservation of olfactory preference, we tested two additional, independent genetic reagents alongside our original RNAi line. The dSetdb1 gene (also known as *eggless*, FBgn0086908) is located on chromosome 2 R at position 24,775,534..24,779,901. The three tools disrupt this gene through distinct mechanisms. The first line, dSetdb1 (TRiP. JF01310), expresses a long dsRNA hairpin that targets a broad region within exon 8 of the dSetdb1 transcript. The second dSetdb1 short (TRiP.HMS00112) utilizes a short-hairpin RNA to target a distinct, non-overlapping sequence. Finally, we used the loss-of-function allele, dSetdb1 lof (*egg$^{235}$*). We assessed olfactory behavior in 45-day-old flies, an age at which RNAi control animals exhibit a significant decline in olfactory performance. As expected, aged control flies showed a reduced response, with a mean Preference Index of 0.31 for the attractive odorant 2,3-butanedione and –0.28 for the aversive odorant 3-octanol. In striking contrast, all three distinct methods of disrupting dSetdb1 produced a consistent preservation of olfactory function in aged flies. Compared to aged controls, flies with reduced dSetdb1 expression displayed a significantly stronger attraction to 2,3-butanedione (mean PI of 0.65 for dSetdb1, $p$=0.0006; 0.68 for dSetdb1 short, $p$=0.0003; and 0.67 for dSetdb1 lof, $p$=0.0005). Similarly, their aversion to 3-octanol was significantly enhanced (mean PI of –0.68 for dSetdb1, $p$<0.0001; –0.69 for dSetdb1 short, $p$<0.0001; and –0.63 for dSetdb1 lof, $p$=0.0002) (***Figure 4—figure supplement 1***). This result across three independent genetic tools provides strong evidence that the observed preservation of olfactory function is a specific consequence of disrupting dSetdb1 and not an off-target artifact.

## Epigenetic modulation of the UPR$^{MT}$ influences olfactory function in an OPN-cell autonomous manner

As olfactory function is a complex behavior dependent on multiple central and peripheral neuronal populations, we investigated whether age-related changes in H3K9 methylation, UPR$^{MT}$, and olfactory function were specifically associated with cholinergic OPNs. We first assessed age-dependent changes in H3K9me3 trimethylation in olfactory projection neurons (OPNs). To this end, we selectively label the membrane of OPNs by expressing the CD8::GFP fusion protein using the OPN-specific driver GH146-Gal4. We then performed immunofluorescence to evaluate H3K9me3 levels specifically

in ToPro3-positive nuclei located in GFP-positive neurons (*Figure 1—figure supplement 1D*). Using this method, we observed an increase in trimethylation in aged OPNs compared to young ones (*Figure 4A–C*). We then evaluated the regulation of dSetdb1 in aging-associated OPNs trimethylation by generating flies carrying the knockdown of dSetdb1 specifically in cholinergic OPNs. Remarkably, H3K9 trimethylation levels in aged flies with OPN-specific dSetdb1 knockdown were not significantly different from young control flies (*Figure 4A–C*).

To further explore the neuronal specificity of the UPR$^{MT}$ effect, we assessed Hsp60::dsRed reporter activity specifically in aged OPNs. The response of the UPR$^{MT}$ sensor in CD8::GFP tagged neurons increased in young flies treated with PQ compared to animals treated with vehicle. However, this response to the mitochondrial stressor was diminished in aged animals (*Figure 4D–E*). Importantly, dSetdb1 knockdown in OPNs increased reporter activity in response to PQ in aged flies (*Figure 4D–E*). To corroborate these findings, we analyzed a second UPR$^{MT}$ reporter, Hsc70-5::dsRed. In young control flies (0 dpe), PQ treatment significantly induced reporter expression compared to vehicle-treated animals (*Figure 4—figure supplement 2A–B*). This response was lost with age, as reporter levels in PQ-treated aged flies (45 dpe) were significantly lower than in their young counterparts and were not significantly different from vehicle-treated aged flies. OPN-specific knockdown of dSetdb1 maintained the response to PQ in aged flies, resulting in significantly higher reporter activation compared to PQ-treated aged controls. Furthermore, dSetdb1 knockdown also elevated the basal expression of Hsc70-5::dsRed in aged flies under vehicle conditions. These findings demonstrate that dSetdb1 knockdown preserves the transcriptional response of UPR$^{MT}$ signaling in aged neurons and suggest that H3K9me3-mediated repression acts as an epigenetic brake on mitochondrial stress responses during aging.

Single-cell expression analysis specifically in cholinergic OPNs using Scope revealed a decrease in the UPR$^{MT}$-associated chaperones Hsp60 and Hsc70-5 in aged OPNs (*Figure 3F–H*), with constant levels of dSetdb1 and lower levels of Kdm2 (*Figure 4I–J*). Importantly, this data suggests that the observed increase in trimethylation levels within OPNs may be associated with a decline in demethylase activity as flies age. Underlying significant implications for the regulation of UPR$^{MT}$ and the overall epigenetic landscape in aged neurons.

Having demonstrated that dSetdb1 is essential for the increase in H3K9me3 in aged flies, preventing the activation of UPR$^{MT}$ specifically in OPNs, we next evaluated olfactory function. Knockdown of dSetdb1 only in cholinergic OPNs improved olfactory function in aged flies compared to controls. We then assessed if this improvement in olfactory function was dependent on UPR$^{MT}$. To this end, dve, or crc were knocked down specifically in cholinergic OPNs in dSetdb1-deficient flies. Notably, knockdown of dve or crc suppressed the maintenance of olfactory function induced by dSetdb1 knockdown, mirroring the olfactory capacity of control aged flies (*Figure 4L*). Additionally, downregulation of dve and crc only in OPNs reduced olfactory performance in young flies to levels comparable to that of aged control flies.

## Downregulation of dSetdb1 in OPNs restores age-associated mitochondrial morphological abnormalities and reduces mROS levels

As changes in UPR$^{MT}$ activation can influence mitochondrial morphology and function, potentially triggering degenerative mechanisms, we investigated mitochondrial morphology in OPNs by expressing a mitochondrially targeted GFP. We examined mitochondrial morphology in three distinct compartments of OPNs, including cell bodies in the AL, the axonal tract, and the presynaptic terminal-enriched lateral horn (LH, *Figure 5A*). In the AL of aged flies, a marked decrease in total mitochondrial volume was observed, along with increased mitochondrial fragmentation and sphericity. Notably, the targeted downregulation of dSetdb1 within OPNs mitigated these age-related changes, resembling the values observed in young control flies. Surprisingly, dSetdb1 knockdown also resulted in reduced mitochondrial fragmentation in young flies, suggesting a potential disruption of the mitochondrial network when compared with age-matched controls (*Figure 5B–F*). Within the axonal tract, aged control axons exhibited a significant reduction in mitochondrial volume when compared to younger flies (*Figure 5G–H*). Targeted knockdown of dSetdb1 effectively maintained mitochondrial volume. Lastly, no significant changes in mitochondrial parameters were observed in the LH of aged flies for both genotypes (*Figure 5I-J*, *Figure 5—figure supplement 2*).

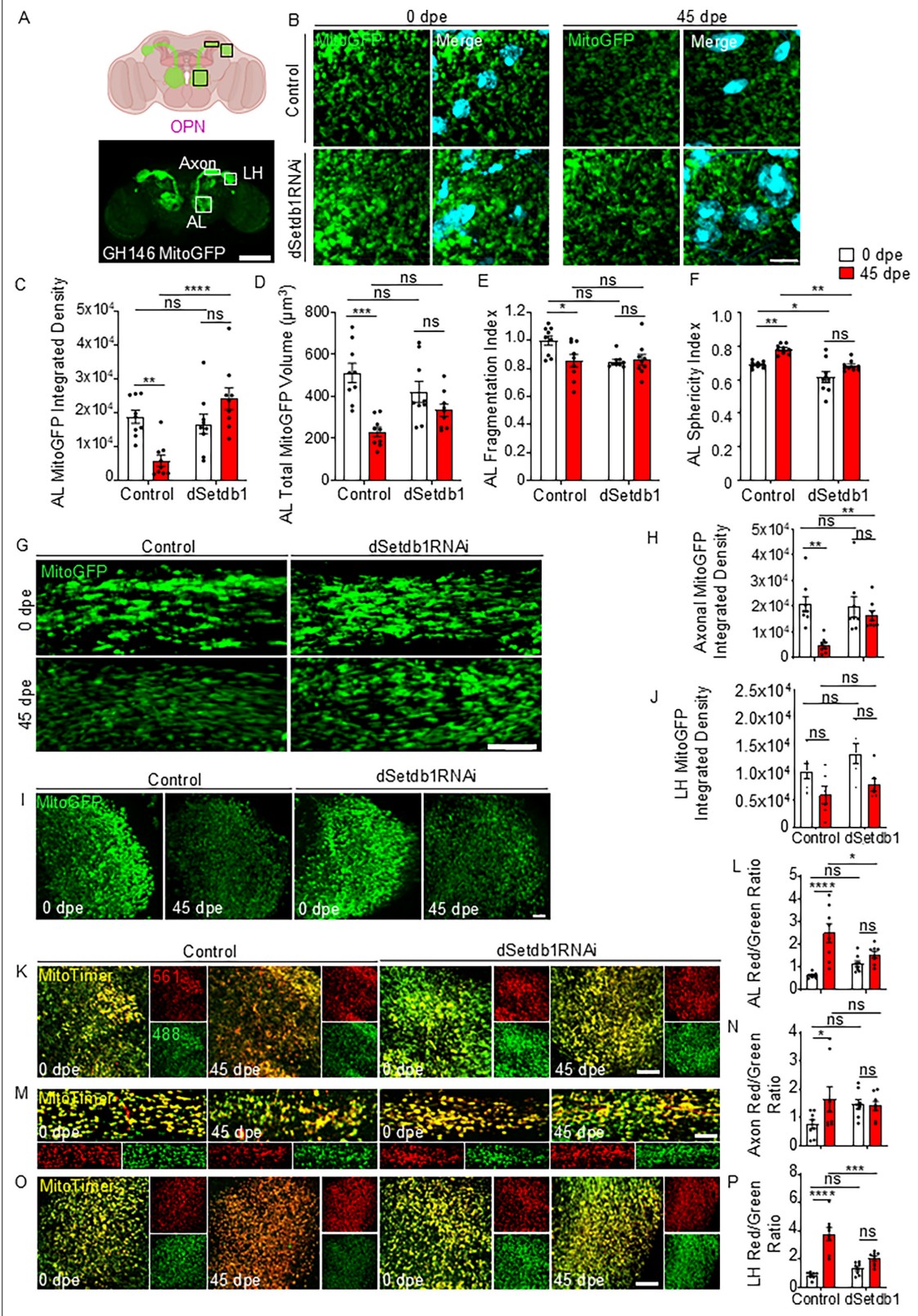

**Figure 5.** dSetdb1 downregulation mitigates age-related mitochondrial oxidation and preserves mitochondrial morphology in OPNs. (**A**) Scheme of GH146-driven GFP expression in olfactory projection neurons (OPNs), and mitochondrial GFP reporter expressed in those neurons. Neurons possess their soma in the antennal lobe (AL) and project their axons to the synapse-enriched zone, the lateral horn (LH). Scale bar, 100 µm. (**B**) Mitochondria labeled with GFP in the AL of OPNs of 0 and 45 days post eclosion (dpe) flies bearing the dSetdb1 RNAi; mitochondria: green, nuclei: cyan. Scale bar,

*Figure 5 continued on next page*

*Figure 5 continued*

5 µm. (**C**) Mitochondrial integrated density in the AL of 0 and 45 dpe flies with dSetdb1 knockdown analyzed via Two-way ANOVA with Bonferroni's multiple comparisons test. Control flies showed a significant difference at 0 vs 45 dpe ($p=0.0018$, n=9), while dSetdb1 RNAi flies showed no significant change ($p=0.0855$, n=9). (**D**) Analysis of total mitochondrial volume in the AL of OPNs with dSetdb1 knockdown compared to controls at 0 and 45 dpe. Two-way ANOVA with Bonferroni's multiple comparisons test indicates a decrease in mitochondrial volume in control flies from 0 to 45 dpe ($p<0.0001$, n=9). dSetdb1 RNAi of 0 vs 45 dpe flies did not show a change in volume ($p=0.2387$, n=9). (**E**) AL mitochondrial fragmentation index of images shown in B. Two-way ANOVA Bonferroni's multiple comparisons test for mitochondrial fragmentation index in control flies from 0 to 45 dpe ($p=0.0145$, n=9) and dSetdb1 RNAi flies showed no significant change ($p>0.9999$, n=9). Control vs. dSetdb1 RNAi at 0 dpe ($p=0.0101$, n=9); no significant change at 45 dpe ($p>0.9999$, n=9). (**F**) AL sphericity index of mitochondria from images shown in B. Graph shows results of Two-way ANOVA with Bonferroni's multiple comparisons test. For 0 vs 45 dpe, control flies ($p=0.0019$, n=9) and dSetdb1 RNAi flies ($p=0.0257$, n=9). At 0 dpe, control vs dSetdb1 RNAi flies ($p=0.012$, n=9), and at 45 dpe ($p=0.0008$, n=9). (**G**) Representative images of axonal mitochondria in the green of 0 and 45 dpe flies bearing the knockdown of dSetdb1. Scale bar, 5 µm. (**H**) Axonal MitoGFP integrated density; control increase ($p=0.0005$, n=9), dSetdb1 RNAi ($p=0.754$, n=9). (**I**) LH MitoGFP confocal images. Scale bar, 10 µm. (**J**) LH MitoGFP integrated density; control ($p=0.1197$, n=9), dSetdb1 RNAi ($p=0.0751$, n=9). (**K**) Representative images of 0 and 45 dpe *Drosophila*'s AL showing GH146-GAL4;UAS-MitoTimer, an in vivo mitochondrial oxidation reporter. Reduced (green) and oxidized (red) MitoTimer signals are shown, and merge (yellow). Scale bar, 5 µm. (**L**) AL Red/Green integrated density ratio; 0 vs 45 comparison reports a significant oxidation increase in control flies ($p<0.0001$; n=8), while dSetdb1 RNAi flies show a non-significant change ($p=0.4359$; n=8). Control vs dSetdb1 RNAi at 0 dpe ($p=0.2618$; n=8) and at 45 dpe ($p=0.0143$; n=8). (**M**) Representative images of 0 and 45 dpe *Drosophila*'s axons showing GH146-GAL4;UAS-MitoTimer. Scale bar, 5 µm. (**N**) Axonal Red/Green integrated density ratio; 0 vs 45 comparison reports an increase in oxidation in control flies ($p=0.0447$; n=8), with no change in dSetdb1 RNAi flies ($p>0.9999$; n=8). Control vs dSetdb1 RNAi at 0 dpe ($p=0.1266$; n=8) and at 45 dpe ($p>0.9999$; n=8). (**O**) Representative images of 0 and 45 dpe *Drosophila*'s lateral horn (LH) showing GH146-GAL4;UAS-MitoTimer. Scale bar, 5 µm. (**P**) LH Red/Green integrated density ratio; 0 vs 45 comparison reports a significant oxidation increase in control flies ($p<0.0001$; n=8), while dSetdb1 RNAi flies do not display change ($p=0.1263$; n=8). Control vs dSetdb1 RNAi at 0 dpe ($p=0.4114$; n=8) and at 45 dpe ($p=0.0001$; n=8). P-value: ***$p<0.0001$; ***$p<0.001$; **$p<0.01$; *$p<0.05$; ns $>0.05$. For all quantified panels, each n represents one brain from an individual animal. All error bars represent mean ± SEM.

The online version of this article includes the following figure supplement(s) for figure 5:

**Figure supplement 1.** Quantification of fluorescent signal within GFP-labeled mitochondria expressed in olfactory projection neuron (OPN) in the antennal lobe, axons, and lateral horn.

**Figure supplement 2.** Analysis of mitochondrial morphology in olfactory projection neurons (OPNs) unaffected by dSetdb1 downregulation with aging in *Drosophila*.

We next assessed mitochondrial oxidation levels as a surrogate marker of mitochondrial function (***Morató and Sandi, 2020***; ***Laker et al., 2014***). To this end, we employed the UAS-MitoTimer construct, a mitochondrial oxidation reporter (***Laker et al., 2014***; ***Hernandez et al., 2013***). This tool relies on the expression of a green fluorescent protein that transitions to red fluorescence upon oxidation (***Laker et al., 2014***). Compared to young control flies, we observed an increase in mitochondrial oxidation in older control flies within the AL (***Figure 5K–L***), axonal tract (***Figure 5M–N***), and the LH (***Figure 5O–P***). Remarkably, downregulation of dSetdb1 in OPNs reversed the age-related mitochondrial oxidation in the three analyzed neuronal regions. Our results indicate that the downregulation of dSetdb1 in OPNs activates the UPR^MT during aging, a process that reverses age-associated changes in mitochondrial morphology and effectively prevents the accumulation of mitochondrial oxidation in vivo.

## Epigenetic regulation of UPR^MT by dSetdb1 modulates age-dependent neurodegeneration of OPNs

Loss of neuronal function is often linked to degenerative structural changes in the neuronal circuit (***Jellinger, 2010***; ***Levenson et al., 2014***), a phenotype extensively associated with mitochondrial dysfunction. Therefore, we investigated whether UPR^MT activation regulates neuronal integrity throughout aging in OPNs. To assess the impact of UPR^MT modulation on neuronal integrity, we first assessed changes in neuronal number throughout aging by counting nuclei from GFP-positive OPNs. In control flies, aging resulted in a significant decrease in the number of OPNs. Remarkably, this age-related neuronal loss was prevented in flies where dSetdb1 was downregulated only in OPNs (***Figure 6A–B***). We next evaluated axonal integrity in GFP-labeled OPNs. In aged control flies, a reduction in axonal integrated density was observed when compared to young control flies, consistent with the previously noted decrease in the total number of neurons. Notably, dSetdb1 knockdown protected against the decline in axonal integrity of OPNs associated with aging (***Figure 6C–D***).

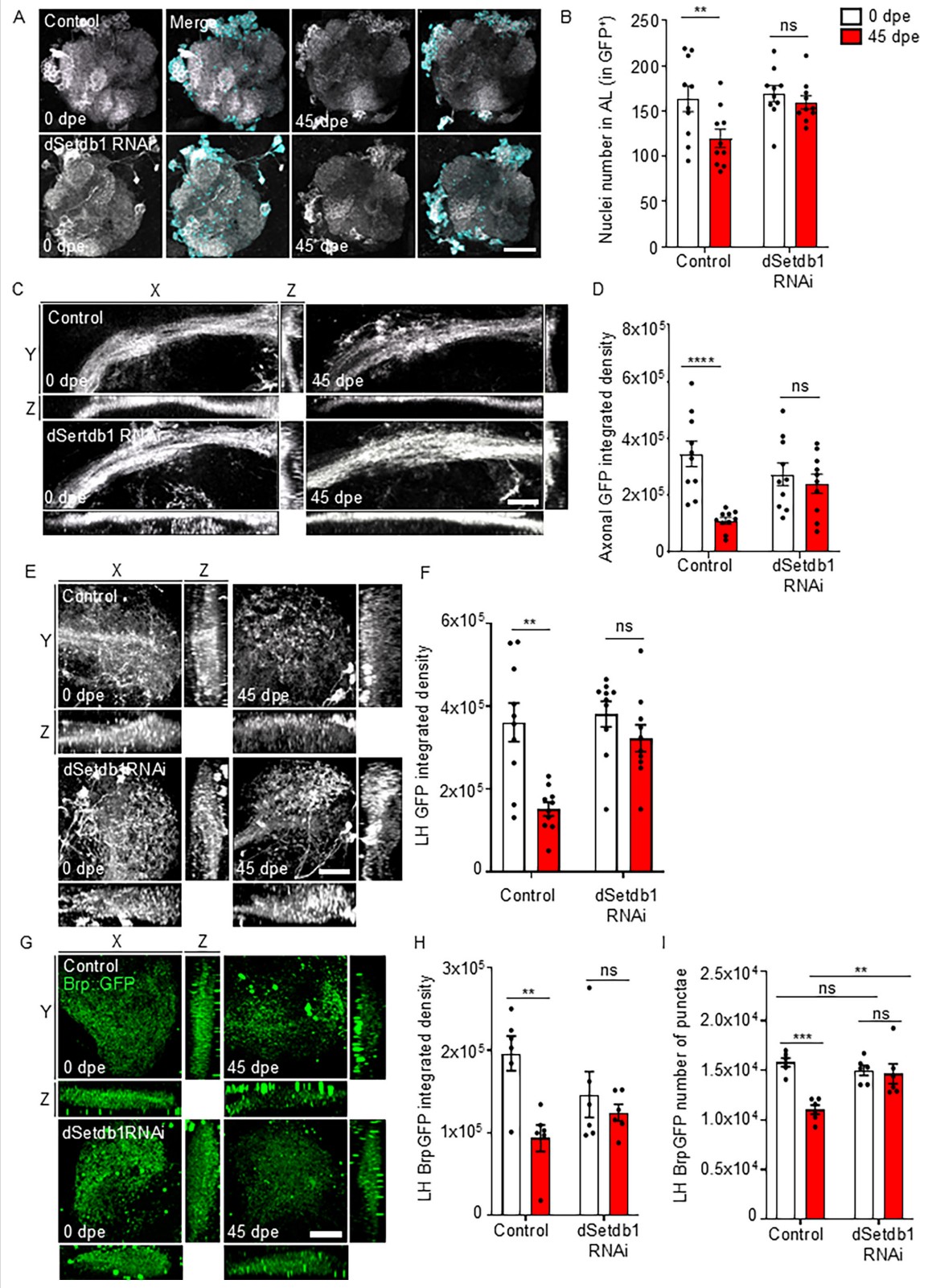

**Figure 6.** dSetdb1 knockdown preserves neuronal integrity and synaptic density in the aging *Drosophila* OPNs. (**A**) GFP-labeled olfactory projection neurons (OPNs) in the antennal lobe (AL) of 0 and 45 days post eclosion (dpe) control flies and flies with knockdown of dSetdb1 and dve. GFP-positive OPNs are gray, and the right panel shows merged channels with nuclei labeled with ToPro3 in cyan. Scale bar, 20 μm. (**B**) Nucleus count in GH146-positive OPNs. Graph and two-way ANOVA with Bonferroni's multiple comparisons between 0 and 45 dpe show that control flies had a significant

*Figure 6 continued on next page*

*Figure 6 continued*

decrease in nucleus count ($p$=0.0233, n=10). dSetdb1 RNAi ($p$>0.9999, n=10); Control vs. dSetdb1 RNAi of 45 dpe, ($p$=0.0398, n=10). (**C**) Orthogonal view of the 3D reconstruction of distal axonal tract of OPNs tagged with GFP. Panel shows the combination of axis Y and X, Z and X, and Z and Y for axons from control flies and knockdown flies for dSetdb1. Scale bar, 5 μm. (**D**) Quantification of axonal integrated density of axons shown in C. Two-way ANOVA multiple comparison between Control flies shows a significant decrease in axonal integrated density from 0 (white bars) to 45 dpe (red bars) ($p$<0.0001, n=10). dSetdb1 RNAi ($p$>0.9999, n=10). (**E**) Representative images of orthogonal view of the 3D reconstruction of GFP-tagged OPNs in the LH. Images show the combination of axis Y and X, Z and X, and Z and Y. Panel shows LH from 0 and 45 dpe control flies, knockdown flies for dSetdb1 and dve. Scale bar, 10 μm. (**F**) Quantification of GFP integrated density in the LH of images shown in E. Two-way ANOVA with Bonferroni's multiple comparisons test shows a significant decrease in GFP volume in the LH of control flies from 0 to 45 dpe ($p$<0.0001, n=10), and no change for dSetdb1 RNAi ($p$>0.9999, n=10). (**G**) Orthogonal view of representative 3D reconstruction images of Brp::GFP-labeled presynaptic densities in LH of 0 and 45 dpe flies bearing the dSetdb1 GH146 knockdown. Scale bar, 20 μm (**H**) Quantification of BrpGFP integrated density of images shown in G. LH Brp::GFP integrated density showed a significant decrease in Brp::GFP integrated density in control flies from 0 to 45 dpe ($p$=0.0031, n=6), while dSetdb1 RNAi flies did not show a significant change ($p$=0.0758, n=6). (**I**) Quantification of the number of presynaptic densities labeled with BrpGFP in the LH of flies bearing the dSetdb1 knockdown shown in G. Control flies showed a reduction in the number of presynaptic densities labeled with Brp::GFP from 0 to 45 dpe ($p$<0.0001, n=6), and dSetdb1 RNAi showed no significant change ($p$>0.9999, n=6). White and red bars represent 0 and 45 dpe flies, respectively. n=independent fly brain. P-value: ****$p$<0.0001; ***$p$<0.001; **$p$<0.01, *$p$<0.05 and ns >0.05. For all quantified panels, each n represents one brain from an individual animal. All error bars represent mean ± SEM.

It has been previously demonstrated that the age-associated decline in olfactory function in *Drosophila* is associated with the loss of synapses in OPNs (*Hussain et al., 2018*). Thus, we focused on the LH, a region enriched in presynaptic connections of OPN neurons (*Hussain et al., 2018*; *Das Chakraborty and Sachse, 2021*). Control flies exhibited a significant decrease in LH integrated density throughout aging, which was prevented by downregulation of Setdb1 in OPNs (*Figure 6E–F*). This data suggests that H3K9-dependent UPR^MT activation plays a crucial role in maintaining neuronal integrity within this presynaptic enriched region. To gain a more detailed insight into synaptic zones, we employed the Brp::GFP reporter, a fusion protein that specifically accumulates in presynaptic buttons, facilitating visualization and quantification of presynaptic puncta (*Mosca and Luo, 2014*). Our analysis revealed that both total volume and number of GFP puncta in the LH of aged control flies were reduced compared to their younger counterparts (*Figure 6G–I*). When dSetdb1 was downregulated in OPNs, we observed a decrease in this age-dependent integrated density decline. Notably, there was no significant difference between the number of Brp::GFP-positive puncta in young and aged dSetdb1 knockdown flies (*Figure 6I*).

These findings collectively demonstrate that the targeted downregulation of dSetdb1 plays a causal role in preserving neuronal numbers and axonal integrity in OPNs. This intervention not only maintains presynaptic densities but also actively contributes to the demethylation of H3K9 and the activation of UPR^MT, which are integral to the preservation of olfactory function during aging (*Figure 7*). By modulating these key processes, our results establish a direct link between the epigenetic regulation by dSetdb1 and the mitigation of age-related neurodegeneration in OPNs, underscoring the potential of targeted epigenetic interventions in maintaining neural health in the olfactory system.

## Discussion

The UPR^MT plays a critical role in preserving mitochondrial homeostasis (*Muñoz-Carvajal and Sanhueza, 2020*). Therefore, any change in its activation potential, whether caused by physiological or pathological factors, could impact mitochondrial function (*Zhou et al., 2022*; *Merkwirth et al., 2016*; *Yuan et al., 2020*; *Ng et al., 2021*). Although it is widely accepted that olfactory function declines with age, diverse factors have been associated with this age-related impairment (*Stevens et al., 1989*; *Wang et al., 2017*; *Xu et al., 2020*; *Dweck et al., 2018*; *Cerf-Ducastel and Murphy, 2003*). Our findings indicate that the age-dependent decline of olfactory function in *Drosophila* is associated with a decrease in the activation capacity of the UPR^MT in olfactory neurons. Crucially, the reduction in UPR^MT activation is linked to an increase in H3K9me3, which is dependent on the methylation activity of dSetdb1. Importantly, targeting this methylation process can effectively prevent age-related neuronal degeneration and restore the loss of olfactory function associated with aging.

Our study uncovers a novel aspect of epigenetic regulation in the aging process, emphasizing the specific role of dSetdb1 in modulating H3K9me3 levels and suppressing the UPR^MT. While previous research has established the impact of H3K9 methylation on the UPR^MT, primarily through the actions

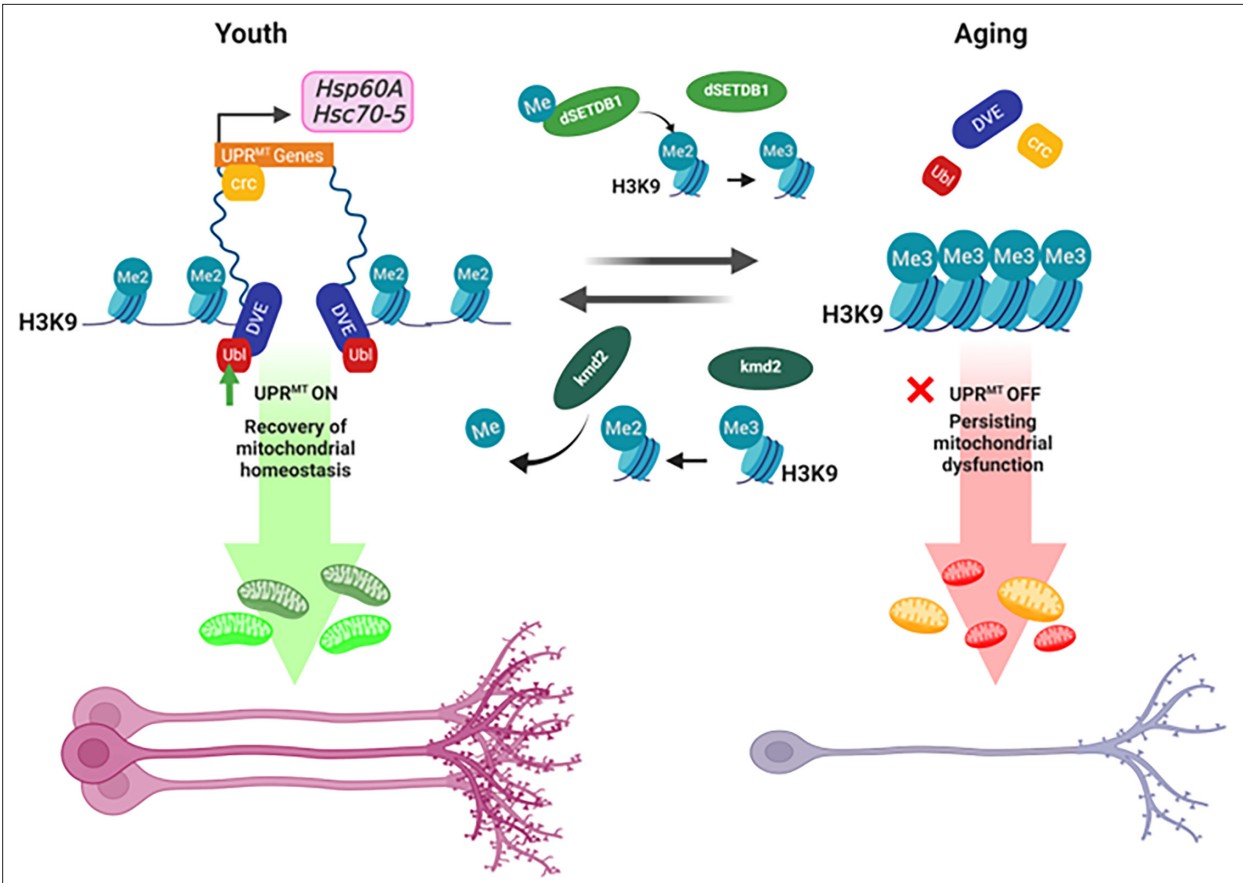

**Figure 7.** Schematic representation of age-related changes in H3K9 methylation and its impact on UPR^MT and neuronal integrity in *Drosophila*. In young organisms, mitochondria are challenged by various insults that lead to the accumulation of mitochondrial dysfunction, causing damage and activating a retrograde response from mitochondria to the nucleus. Ubl, crc, and DVE are translocated to the nucleus, and DVE maintains an open chromatin state, allowing the binding of transcriptional modulators of the UPR^MT. This event activates the transcription of chaperones and proteases to recover mitochondrial homeostasis and oxidation. During aging, trimethylation of H3K9 is a mark of heterochromatin associated with the repression of transcription. This trimethylated state of H3K9 does not allow the binding of the transcriptional modulators of UPR^MT, dve, crc, and ubl, inhibiting the mitochondrial response to aging-causing damage. Thus, mitochondrial function persists and builds up in a time-dependent manner, increasing mitochondrial oxidation and contributing to aging phenotypes, such as neurodegeneration marked by the reduction of OPNs, axonal volume, and presynaptic connections. Figure created with BioRender.com.

of the demethylases JMJD 3.1 and JMJD 1.3, as well as methyltransferases, such as MET-2 (dSetdb1 orthologue), SET6, BAZ2 in worms and mice, respectively (*Tian et al., 2016*; *Merkwirth et al., 2016*; *Yuan et al., 2020*), the specific function of Setdb1 in this context remained unclear. Our analysis indicates that in the AL of *Drosophila* brain, the absence of dSetdb1, also known as *eggless*, leads to a reduction in H3K9me3 in aged flies, suggesting its role as a tri-methyltransferase that acts as an epigenetic modulator of UPR^MT during aging. The specificity of this effect was rigorously confirmed, as two independent RNAi lines and a classic loss-of-function allele for dSetdb1 (egg^235) all consistently rescued the age-related decline in olfactory behavior (*Figure 4—figure supplement 1*). This finding aligns with evolutionary conservation in epigenetic regulation across species. Supporting this, research shows that in *C. elegans*, knocking out met-2 also reduces H3K9me3, suggesting a conserved mechanism (*Delaney et al., 2022*). However, previous research on the role of met-2 in UPR^MT regulation only focused on H3K9Me2 and did not extend their examination to H3K9me3 (*Tian et al., 2016*). Additionally, the role of SET6 in tri-methylating H3K9 and reducing UPR^MT in mice further highlights a possible conserved epigenetic pathway influencing aging across distinct species (*Yuan et al., 2020*).

Existing research underscores the beneficial role of UPR^MT in maintaining cellular integrity, primarily through maintaining mitochondrial function, a critical factor for healthy aging (*Durieux et al., 2011*; *Mouchiroud et al., 2013*; *Dillin et al., 2002*). Indeed, the overexpression of the histone demethylases

JMJD-1.2 and JMJD-3.1 extends lifespan in *C. elegans* (*Merkwirth et al., 2016*). Conversely, reduced expression of UPR^MT nuclear effectors ATFS-1, UBL-5, and DVE-1, as well as demethylases JMJD-1.2 and JMJD-3.1, compromises lifespan and cellular viability (*Merkwirth et al., 2016*; *Durieux et al., 2011*; *Houtkooper et al., 2013*; *Cooper et al., 2017*; *Lan et al., 2019*). Recent work reveals the fine-tuned regulation of UPR^MT along aging, particularly through the H3K9 methyltransferase SET-6 and the epigenetic reader BAZ-2, which modulate gene expression related to mitochondrial health and stress responses, essential for neuronal viability (*Yuan et al., 2020*). Importantly, beyond the impact on lifespan, UPR^MT regulation has profound implications for neuronal function. In *C. elegans*, mitochondrial function was found to influence pharyngeal pumping (eating) and defecation rates, crucial for lifespan (*Dillin et al., 2002*). Additionally, deficits in dopamine-dependent behaviors were observed in pdr-1 and pink-1 mutants, indicative of neuronal dysfunction without neuronal loss. This dysfunction is exacerbated by the downregulation of atfs-1, which is critical for UPR^MT (*Cooper et al., 2017*). In a mammalian context, Baz2b ablation enhanced mitochondrial function in the hippocampus and cerebellum in older mice, suggesting its role in modulating age-related cognitive decline (*Yuan et al., 2020*). This was accompanied by improved performance in locomotion, reflecting preserved motor functions in aged mice. Beyond these, UPR^MT regulates hippocampal neural stem cell aging, with implications for cognitive functions (*Wang et al., 2023b*) and affects skeletal muscle aging, as exercise improves coordination of UPR^MT and mitophagy in aging skeletal (*Wang et al., 2023a*). Moreover, it plays a critical role in fertility and reproductive aging (*Seli et al., 2019*), suggesting that UPR^MT disruption can pivot healthy aging towards pathological states. Our data indicate that reducing H3K9me3 levels during aging to enhance UPR^MT activation is beneficial for olfactory function and supports the prevailing hypothesis that modulation of epigenetic regulators, which suppress UPR^MT transcriptional activation, constitutes a viable therapeutic approach for ameliorating mitochondrial dysfunction associated with aging.

While the precise molecular mechanisms of the UPR^MT in *Drosophila* may differ from those in *C. elegans*, our findings demonstrate the critical role of this pathway in maintaining olfactory function during aging. The rescue of age-related olfactory decline by dSetdb1 knockdown, which is abolished by further knockdown of the UPR^MT transcriptional activators dve and crc, highlights the importance of this pathway in neuronal maintenance. The *Drosophila* homolog of ATF5 and ATF4, crc, appears to play a key role in this process. Although its direct translocation from the mitochondria to the nucleus remains to be definitively shown, crc is known to be a target of mitochondrial stress and the UPR^ER (*Ryoo, 2015*; *Vasudevan et al., 2022*; *Ryoo et al., 2007*). Moreover, previous studies have demonstrated that mitochondrial dysfunction in *Drosophila* neurons can lead to the activation of crc via the UPR^ER (*Hunt et al., 2019*), suggesting potential crosstalk between these stress response pathways. This complex interplay between the UPR^MT and other cellular stress responses, including the ISR, has also been observed in mammals (*Jenkins et al., 2021*; *Quirós et al., 2017*; *Anderson and Haynes, 2020*). The involvement of ATF4, a key ISR effector, in the mammalian UPR^MT further supports the idea of interconnected stress signaling networks (*Quirós et al., 2017*; *Jiang et al., 2020*).

All together, these findings suggest that crc may function as a critical node integrating signals from both the ER and mitochondria to orchestrate a coordinated cellular response to stress. Further investigation is warranted to fully elucidate the molecular mechanisms by which crc regulates the UPR^MT and its potential role in mediating crosstalk between different stress response pathways in *Drosophila*.

Brain aging exhibits distinct regional variations across multiple levels, including gene expression, organelle, and neuronal function (*Burtscher et al., 2015*; *Almanzar et al., 2020*; *Aibar et al., 2017*). Our analysis of gene expression from single-cell studies reveals neuronal-specific transcriptional expression for H3K9-regulating enzymes, such as dSetdb1, utx, and kdm2 in aged vAChT, vGlut, and Gad1 neurons. It has been previously demonstrated that H3K9me3 levels are not uniform across the brain, varying based on brain regions or neuronal types (*Cao and Dang, 2018*; *Benayoun et al., 2015*; *Kane and Sinclair, 2019*). Neurodegenerative conditions, characterized by the selective degeneration of specific neurons and their projections, exhibit differential neuronal vulnerability that is intricately linked to variations in neuronal morphology, activity patterns, and gene expression profiles within these affected structures (*Fu et al., 2018*; *Paß et al., 2021*; *Morrison et al., 1998*). Our data suggest that the age-dependent reduction in H3K9 demethylation enzymes within specific neuronal populations may contribute to differential neuronal vulnerability along aging, which deserves further exploration.

The increase we had shown in the levels of H3K9me3 in the brains of aged fruit flies aligns with prior studies reporting a rise in H3K9me3 in aged *Drosophila* heads (*Herz et al., 2010*). Studies performed in *C. elegans* have revealed that as age progresses, there is an increase in the expression of the H3K9me3 methyltransferase SET-6 and the epigenetic reader BAZ-2. Remarkably, inhibiting their expression has been linked to preservation of pharyngeal pumping in these organisms (*Yuan et al., 2020*). In mice, administering an inhibitor for the histone methyltransferase SUV39H1, which is responsible for the trimethylation of H3K9, was found to mitigate age-associated cognitive decline and augment dendritic spines in the hippocampus (*Snigdha et al., 2016*). Such findings support the notion that reducing H3K9me3 levels might enhance functionality of different brain modules during aging. While our findings suggest that reducing hypermethylation in aging could potentially enhance UPR$^{MT}$ response, it is critical to acknowledge that methylation processes also govern a vast of other cellular and neuronal functions (*Richard et al., 2010*). For instance, histone methylation plays a crucial role in gene expression regulation, cellular differentiation, and even neuronal activity (*Park et al., 2022*; *Basavarajappa and Subbanna, 2021*), all of which could be inadvertently impacted by broad-spectrum epigenetic interventions. This pleiotropic nature of methylation underscores the importance of a targeted approach.

Mitochondria play a central role as primary sensors for degenerative stimuli (*Court and Coleman, 2012*). With aging, neurodegeneration is often preceded by mitochondrial dysfunction, which manifests as morphological changes, including swelling, fragmentation, reduced volume, and increased oxidative stress (*Stahon et al., 2016*; *Olesen et al., 2020*; *Venkateshappa et al., 2012*; *Wang et al., 2021*; *Rottenberg and Hoek, 2017*; *Rottenberg and Hoek, 2021*; *Salvadores et al., 2017*; *Kim et al., 2018*; *Arrázola et al., 2019*). Consistent with prior studies in aged flies (*Hussain et al., 2018*), we did not observe an increase in fragmentation within axons of the OPNs and lateral horn (LH), which could be attributed to the specific neuronal types under investigation (*Burman et al., 2012*). Interestingly, our observations align with those from other studies (*Hussain et al., 2018*), as we identified an age-related deterioration in *Drosophila's* olfactory circuits, which coincides with a rise in oxidative mitochondria. Current research underscores the central role of UPR$^{MT}$ activation in orchestrating mitochondrial morphology and function (*Zhang et al., 2016*; *Liu et al., 2020*; *Chen et al., 2021*; *Yan et al., 2021*; *Wang et al., 2019*). Importantly, our study demonstrated that genetic inhibition of dSetdb1 restored youthful levels of H3K9me3, enabling UPR$^{MT}$ activation to restore mitochondrial morphology and oxidative status. This maintenance of cellular viability through UPR$^{MT}$ activation parallels findings from a recent study, which revealed that mild mitochondrial dysfunction-dependent UPR$^{MT}$ activation protects cardiomyocytes against cardiac ischemia-reperfusion injury in mice (*Bomer et al., 2021*). Also, NAD +activation of the UPR$^{MT}$ rejuvenated muscle stem cells in aged mice (*Zhang et al., 2016*), highlighting the role of UPR$^{MT}$ in altering aging markers in stem cells and extending lifespan.

Our research highlights the crucial role of UPR$^{MT}$ regulation in the age-related decline of olfactory function. Focusing on the olfactory system in aged *Drosophila*, we have demonstrated detrimental effects of epigenetic changes on mitochondrial function, impacting neuronal survival. The importance of olfaction extends beyond sensory perception, with human olfactory processing is intricately linked to emotions and memories, mediated by the limbic system and cerebral cortex (*Churchwell and Yurgelun-Todd, 2013*; *Dan et al., 2021*). A compromised sense of smell is not only associated with depression in a significant number of cases (*Croy et al., 2014*) but also frequently precedes the onset of age-related neurodegenerative diseases, such as Alzheimer's (*Vasavada et al., 2015*; *Zou et al., 2016*) and Parkinson's disease (*Leonhardt et al., 2019*; *Cecchini et al., 2019*). Our findings underscore the need for further exploration of the UPR$^{MT}$ pathway and its epigenetic regulation as a potential target for developing interventions to mitigate the decline in neuronal function associated with the aging process.

## Materials and methods
### *Drosophila* strains and culture

Strains carrying the following transgenes were obtained from the Bloomington *Drosophila* Stock Center (BDSC) from Indiana University: UAS-Mito-GFP (BL#8443, encodes the 31 amino acid mitochondrial import sequence from human cytochrome C oxidase subunit VIII fused to the N-terminus of the Green fluorescent protein), UAS-MitoTimer (BL#57323, encodes a mitochondrial targeting

sequence with a roGFP which turns its fluorescence from green to red when oxidized), Elav-Gal4 -Gal4 (BL#485), Actin-Gal4 (BL#9431), GH146-Gal4 (BL#30026), GH146,GFP (BL#36500); RNAi lines for UPR$^{MT}$ genes,including crc (BL# 25985), dve (BL#26225), ubl (BL#65893), and RNAi lines for UPR$^{MT}$ associated H3K9 methylation enzymes dSetdb1 (TRiP.JF01310, BL# 31352), an independent short-hairpin RNAi line for dSetdb1 (TRiP.HMS00112, BL# 34803), dSetdb1 loss of function (egg$^{235}$, BL#30566), Utx (BL#34076), and Kdm2 (BL#33699). Wild-type flies used as controls are Canton S (BL#64349), and RNAi Control flies from the Transgenic RNA Interference Project (TRIP) (BL#35787). Only female flies were used for all experiments to avoid genetic variation and aggressive behavior from males. All fly stocks were maintained on a standard *Drosophila* medium which consisted of 112.5 g of molasses, 35 g of dry yeast, 90 g of corn flour, 9 g of agar, 2.5 g of Tegosept diluted in 10 ml of ethanol 95%, and 6 ml of propionic acid per 1 L of water; at 25 °C and under a circadian cycle of 12 hr of light and 12 hr of darkness.

## Hsp60::dsRed and Hsc70-5::dsRed reporter generation

To monitor UPR$^{MT}$ transcriptional output in vivo, we generated site-specific transgenic reporters expressing dsRed under regulatory sequences from Hsp60A or Hsc70-5. For the Hsp60A reporter, a 1583 bp genomic fragment corresponding to the 5′ region/5′UTR upstream sequence of Hsp60A was PCR-amplified from w$^{1118}$ genomic DNA (Forward: ACACATTAAGGTTAGGAAGTTCGGA; Reverse: AAGCGAAACTGGCAAACG) and combined with a dsRed-Stop fragment amplified from a 3xP3-dsRed plasmid template (Forward: ATGGCCTCCTCCGAGGAC; Reverse: CTACAGGAACAGGTGG TGGC). These fragments were assembled into pAttB between HindIII/KpnI sites using a sequence and ligation-independent method. For the Hsc70-5 reporter, a 293 bp 5′ upstream/5′UTR fragment was PCR-amplified from w$^{1118}$ genomic DNA (Forward: GTTTTCAAACCACCTTGTGC; Reverse: CAGA AACTTGGGTACGCG) and assembled upstream of dsRed-Stop in pAttB using the same strategy. Both constructs were integrated at the attP2 (68A4) landing site by PhiC31-mediated transgenesis (injection strain: y[1] M{vas-int.Dm}ZH-2A w[*]; P{y[+t7.7]=CaryP}attP2). WellGenetics validated plasmids, microinjected embryos (225 for Hsp60A::dsRed; 208 for Hsc70-5::dsRed), screened transformants by visible marker, and established balanced stocks. Because these are transcriptional reporters, dsRed fluorescence is expected to be predominantly cytosolic, and reporter induction is interpreted as changes in transcriptional output from the corresponding regulatory region rather than mitochondrial targeting.

## Treatment with mitochondrial stressor paraquat and doxycycline

Experiments requiring mitochondrial stressor paraquat (Sigma, 36541) or doxycycline (Sigma, D3447) were performed by supplementing standard *Drosophila* medium with 100 µl of paraquat diluted in dH2O at 10 µM or doxycycline at 100 µM. After PQ addition, vials must be airdried before use. Groups of flies were exposed to paraquat-supplemented medium for 48 hr before experiments were conducted.

## Olfactory functional assay

Olfactory T-maze was used to perform the olfactory behavioral test based on *Hussain et al., 2018*. Briefly, 15 flies are presented with an abrasive odor 0.1 M of Hydroxychloroquine, 3-octanol (Sigma, W358118), or a pleasant odor of 2.3-butanedione (Sigma, B85307) at the end of one arm of the T-maze and at the end of the opposite arm flies are exposed to control solution (vehicle only). Flies have 60 s to discriminate between odors and go to an arm of the T-maze. At the end of the 60 s, an image is acquired, and flies in both arms are counted. The olfactory preference index consists of ((Flies in Experimental Odor – Flies in Vehicle Odor)/(Total flies in the experiment)). The olfactory preference index is calculated for every trial, and every n in the graph corresponds to the mean of five trials of 15 flies each. For statistical analysis when comparing genotype and treatment, two-way ANOVA tests were performed with analysis of variance and Dunnett multiple comparisons for more than two groups using GraphPad Prism 6. P-value: ****$p<0.0001$; ***$p<0.001$; **$p<0.01$, *$p<0.05$ and ns $>0.05$.

## Protein quantification and western blotting

For immunostaining of H3K9me3, samples were prepared as described previously3. Briefly, samples were frozen in liquid nitrogen and ground to a fine powder using a pestle fitted to a 1.5 ml Eppendorf

Centrifuge tube filled with RIPA buffer, which included 50 nM Tris, 150 mM NaCl, 1 mM EDTA, 0.1% of Nonidet P-40 (NP-40), 0.25% of Sodium Deoxycholate, and 0.02% of sodium azide in ddH20. RIPA buffer was supplemented with phenylmethylsulphonyl fluoride (PMSF – Sigma) and a protease inhibitor cocktail (Sigma, P8340). Homogenized samples were incubated at 4 °C for 1 hr and sonicated by sonicator (QSonica) at 40% of equipment maximal amplitude with three pulses in 1 min. Sonicated samples were centrifuged at 500 g to pellet the debris, and all supernatant was transferred to a new tube and then centrifuged for 14 min at 13,000 g at 4° C. The upper soluble phase was transferred to a new 1.5 ml Eppendorf for membrane and plasma proteins and pellet, and 200 µl of liquid phase was kept for nuclear proteins. The pellet was dissolved in the liquid phase by pipetting. Quantification of samples was performed using the Pierce BCA Protein Assay Kit (Thermo Scientific, 23225) under the manufacturer's instructions. Samples were boiled in SDS sample buffer for 15 min, separated on an SDS-PAGE gel, transferred, and revealed using BioRad TransBlot and ChemiDoc, respectively. Primary antibodies used were rabbit anti-H3K9me3 1:2000 (Abcam, ab8898), and loading control anti-Tubulin 1:1000 (Thermo Fisher, MA1-744). Secondary antibodies were anti-rabbit conjugated with Horseradish Peroxidase (HRP) 1:1000 (Thermo Fisher). The stained membranes were briefly incubated in luminol and scanned using ChemiDoc (BioRad). One biological replicate corresponded to a homogenized solution of 20 fly heads minimum. The normalized H3K9me3 levels were calculated by normalizing the ratio of H3K9me3 and loading control to that of young control samples. The significance of the interaction between genotypes and time was calculated by a two-way ANOVA test with Dunnett multiple comparisons using GraphPad Prism 9. P-value: ****$p<0.0001$; ***$p<0.001$; **$p<0.01$, *$p<0.05$, and ns$>0.05$.

## Dissection of adult *Drosophila* brains and confocal microscopy

Flies of the desired genotype were collected in groups of 15 and placed in vials containing *Drosophila* medium. Flies were anesthetized using a $CO_2$ pad. Using Dumont forceps #5, flies were held from the thorax and dipped in the Sylgard petri dish filled with cold PBS. Brains were isolated by removing the exoskeleton from the fly's head and carefully removing the esophagus and air sacs of the flies' brains as previously described (*Lucke et al., 2020*). *For imaging of endogenous fluorescence signal* (*Figures 1A, B, E, G, 2B, D, 4D, 5K, M, O and 6G*) brains were fixed in 2% paraformaldehyde with 0.1% Triton X-100 (Sigma, T9284) for 20 min and then changed to a 4% paraformaldehyde, 0.1% Triton X-100 solution for 20 more minutes, then washed for 10 min three times in PBS Triton X-100 at 0.1% followed by three quick washes with PBS only. Brains were mounted in VectaShield antifade mounting medium (Vector, H1000) for later visualization in an SP8 confocal microscope. For all experiments, all brains were imaged on the same day. *For immunostaining* (*Figures 4A, 5A, B, G, I and 6A, C, E*), brains were dissected as described previously (*Wu and Luo, 2006*). Brains were fixed for 1 hr at room temperature in 4% PFA with 0.1% Triton X-100, washed three times for 10 min with PBS with 0.1% Triton X-100, and blocked for 1 hr with Normal goat Serum (NGS) (Cell Signaling, 5425 S) at 5% in PBS, 0.1% Triton X-100 and stained overnight at 4 °C with primary and after three washes in PBS, 0.1% Triton X-100 with secondary antibodies using the same conditions. The secondary antibody was washed six times for 10 min each with PBS, 0.1% Triton X-100, and three quick washes in PBS before mounting. Washed brains were placed in a stripe of 15–20 µl of Vectashield antifade mounting medium (Vector, H1000) on cover glass (Deltalab, D10004), and imaging was performed in confocal microscope SP8 using the 63 x objective with digital zoom necessary for desired resolution. Fluorescence intensity for each channel was adjusted using control flies to the point that no saturation was observed, then the same parameters were used for all images. Images of antennal lobe sections of *Drosophila* brains and OPNs were taken at a depth of 10 µm using a Z stack separation of 0.6 µm. Primary antibodies used were anti-GFP (Invitrogen, 1:1000), rabbit anti-H3K9me3 (Abcam, ab8898; 1:500), and secondary antibody donkey anti-Rabbit 555 (Thermo Fisher, 1:1000).

## Image quantification

*For quantification of Hsp60::dsRed, Hsc70-5 reporter, and Xbp1::GFP signal*
We selected a Region of interest (ROI) of 100 um$^2$ around the antennal lobe (*Figure 1—figure supplement 3A*) then 3D reconstruction of labeled structures was performed in Imaris Software. The surface (3D models computed from 3D images by a sequence of pre-processing, segmentation, and connected component labeling steps) of the desired signal was rendered. Then we obtained the

volume of the surface and the intensity signal of the reporter inside the surface (Masked Signal), and the following parameters were quantified. *Integrated density:* This parameter represents a cumulative metric of the fluorescence signal within a specified region, denoting the aggregate of signal intensity and its spatial distribution. It is computed by multiplying the average fluorescence intensity by the volume of the signal-bearing domain, thereby yielding a singular value that encapsulates both the concentration and extent of the fluorescent activity. This approach normalizes variations in ROI size, enabling accurate comparisons across samples (*Figure 1—figure supplement 1B*).

### For quantification of Hsp60::dsRed reporter co-expressed with a CD8::GFP reporter

We selected a volume using the GFP signal of labeled neurons in the antennal lobe of *Drosophila* brains (For *Figure 1G* elav-Gal4,CD8::GFP and for *Figure 4D* GH146-Gal4,CD8::GFP) then 3D reconstruction of labeled structures was performed in Imaris Software. The 3D surface of the desired GFP signal was rendered. Then we obtained the volume of the GFP-labeled surface and the intensity signal of the reporter inside the GFP-labeled surface (masked signal) and quantified the integrated density. This parameter represents a cumulative metric of the fluorescence signal within a specified GFP-labeled region, denoting the aggregate of signal intensity and its spatial distribution. It is computed by multiplying the average fluorescence intensity of the reporter by the volume of the GFP signal-bearing domain, thereby yielding a singular value that encapsulates both the concentration and extent of the fluorescent activity within the GFP-labeled neurons. This approach normalizes the Hsp60::dsRed and Hsc70-5::dsRed signal to the GFP signal, accounting for variations in cell number and volume (*Figure 1—figure supplement 1C*).

### Mitochondrial morphology

Mitochondrial changes during aging in OPNs were analyzed by rendering the surface of mitochondrial reporter mitoGFP expressed specifically in the OPNs. Total mitoGFP volume ($\mu m^3$), mitoGFP puncta number, mitoGFP fragmentation index which corresponds to volume ($\mu m^3$) per area, ($\mu m^2$) mitoGFP sphericity index, mitoGFP average size ($\mu m^3$), and integrated density were analyzed (*Figure 5—figure supplement 1*).

### Mitochondrial oxidation

Mitochondrial oxidation was assessed using MitoTimer, a fluorescent protein that shifts from green to red upon oxidation. The analysis involved processing both the green and red channels and determining their ratio. In this context, oxidized mitochondria appeared red, while healthy mitochondria were green, following the methodology outlined in a previous study (*Laker et al., 2014*; *Hernandez et al., 2013*).

### Nuclei number and H3K9me3 quantification

For quantification of the specific signal in Olfactory Projection Neurons, a surface of GFP-labeled OPNs was rendered, then To-Pro3 labeled nuclei signal was masked and the surface rendered, and finally, the signal for H3K9me3 in the nuclei of OPNs was quantified as described previously (*Hussain et al., 2018*; *Figure 1—figure supplement 1D*).

### Neuronal integrity

Neuronal degeneration of OPNs was analyzed by rendering the surface of GFP-labeled OPNs in AL, the distal part of the axon, and axonal terminals in the LH, Integrated density was calculated to determine the amount of signal intensity.

### Presynaptic puncta quantification

For quantifying presynaptic puncta, we employed the Brp::GFP reporter, a fusion of Brunchpilot and GFP. This reporter accumulates in presynaptic buttons, allowing visualization. After rendering the GFP surface, we applied a mask to the GFP channel and counted the puncta, using a threshold ratio of 300 µm in the masked channel to ensure accuracy as described previously (*Mosca and Luo, 2014*). Data was plotted and analyzed using GraphPad Prism 9 Software. To compare the interaction between

age and genotype/treatment, two-way ANOVA and for more than two groups, analysis of variance with Dunnett multiple comparisons was performed using GraphPad Prism 9. P-value: ****$p<0.0001$; ***$p<0.001$; **$p<0.01$, *$p<0.05$ and ns$>0.05$.

## Single-cell RNA-seq data analysis

To analyze gene expression in different cell populations between the *Drosophila* aging brain, we used SCope (http://scope.aertslab.org) or the 'ScopeLoomR' package in R with the scRNA-seq data 'Aerts_Fly_AdultBrain_Filtered_57 k.loom' under accession code GEO:GSE107451. We compared the AUC (Area Under the Curve) values derived from SCope, which indicate the activity levels of genes under regulons across diverse cellular populations. These values reflect the combined activity of gene sets regulated by specific transcription factors, allowing to infer changes in gene expression. By assessing the AUC values for specific genes of interest, namely Hsp60A, Hsc70-5, dSetdb1, Utx, and Kdm2, we could quantitatively evaluate their activity in their respective regulon within distinct neuronal populations, including cholinergic (vAChT), glutamatergic (vGlut), GABAergic (Gad1), and olfactory projection neurons (OPN). This approach allowed us to quantify a proxy for gene abundance and enable a nuanced understanding of the regulatory mechanisms at play. For a comprehensive understanding of the technical underpinnings and applications of SCope in single-cell transcriptomics, we refer to the work by *Davie et al., 2018*, which established a single-cell transcriptome atlas of the aging *Drosophila* brain. All data was tabulated in R and then plotted using GraphPad Prism 9 for statistical analysis.

## Acknowledgements

This work was supported by grants from the Geroscience Center for Brain Health and Metabolism, FONDAP-1231432, Fondo Nacional de Desarrollo Científico y Tecnológico (FONDECYT) N° 1150766, and Financiamiento Basal para Centros Científicos y Tecnológicos de Excelencia Centro Ciencia and Vida (FB210008) (to FAC), Agencia Nacional de Investigación y Desarrollo (ANID) Subvención a la Instalación en la Academia PAI77180059, and FONDECYT Iniciación N°11200981 (to MS). We thank Dr. Ramón Ramirez for his useful support on image processing with the Imaris software at the Center for Integrative Biology, Universidad Mayor de Chile.

## Additional information

### Funding

| Funder | Grant reference number | Author |
|---|---|---|
| Fondo Nacional de Desarrollo Científico y Tecnológico | 1231432 | Felipe A Court |
| Fondo de Financiamiento de Centros de Investigación en Áreas Prioritarias | 15150012 | Felipe A Court |
| ANID Fondo Subvencion a la Instalacion Academia | PAI77180059 | Mario Sanhueza |
| Fondo Nacional de Desarrollo Científico y Tecnológico | 11200981 | Mario Sanhueza |
| Financiamiento Basal para Centros Científicos y Tecnológicos de Excelencia Centro Ciencia and Vida | FB210008 | Felipe A Court |

The funders had no role in study design, data collection and interpretation, or the decision to submit the work for publication.

## Author contributions
Francisco Muñoz-Carvajal, Conceptualization, Formal analysis, Investigation, Methodology, Writing – original draft, Writing – review and editing; Nicole Sanhueza, Investigation, Methodology; Mario Sanhueza, Conceptualization, Resources, Supervision, Validation, Investigation, Methodology, Writing – original draft, Writing – review and editing; Felipe A Court, Conceptualization, Resources, Supervision, Funding acquisition, Validation, Methodology, Project administration, Writing – review and editing

## Author ORCIDs
Francisco Muñoz-Carvajal ![ORCID] https://orcid.org/0009-0003-6650-0374
Nicole Sanhueza ![ORCID] https://orcid.org/0009-0001-5173-0471
Mario Sanhueza ![ORCID] https://orcid.org/0000-0002-0125-0154
Felipe A Court ![ORCID] https://orcid.org/0000-0002-9394-7601

## Decision letter and Author response
Decision letter https://doi.org/10.7554/eLife.103118.sa1
Author response https://doi.org/10.7554/eLife.103118.sa2

## Additional files

### Supplementary files
MDAR checklist

Source data 1. Raw data and analysis outputs for all the figures.

### Data availability
All data generated or analysed during this study are included in the manuscript and supporting files, source data files have been provided.

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
