## [Editor Report]

This important manuscript supports a model that age-dependent increases in H3K9me3, by dSetdb1, repress the mitochondrial UPR, leading to neuronal degeneration and olfactory decline. The evidence is convincing but a few minor limitations remain, such as a mechanistic link between H3K9me3 and UPRmt repression.

---

## [Decision Letter]

[Editors' note: this paper was reviewed by Review Commons.]

---

## [Author Response]

General Statements

We submit a revision plan for our manuscript "Age-dependent H3K9 trimethylation by dSetdb1 impairs mitochondrial UPR leading to degeneration of olfactory neurons and loss of olfactory function in *Drosophila*" by Muñoz-Carvajal et al. for evaluation.

While previous research has linked changes in methylation levels to UPRMT activation and age-related functional decline, the specific effects in the context of neuronal degeneration have not been previously demonstrated. In this work, we utilized behavioral, molecular, and morphological techniques to demonstrate that in the *Drosophila* brain, an age-dependent increase in H3K9 trimethylation by the methyltransferase dSetdb1 reduces the activation capacity of the UPRMT in olfactory projection neurons. This leads to neurodegeneration and the loss of olfactory function, an age-dependent phenotype conserved in multiple species.

We foresee that these results will be of interest to a wide audience as they connect two hallmarks of aging, mitochondrial dysfunction and epigenetic alterations, and how epigenetic changes can lead to neurodegeneration by regulating homeostatic mitochondrial responses.

We have now received the comments from three reviewers and we are prepared to experimentally approach the issues raised. We thank their criticism and suggestions, as well as their very enthusiastic comments. We are extremely pleased as reviewers recognized the important implication of this work:

“The significance of this work is in establishing the roles of H3K9 methylation and UPR mt in the *Drosophila* olfactory system. The results will draw interest from *Drosophila* geneticists interested in the olfactory system and aging.” (reviewer 1).

“This manuscript addresses important questions in the field of mitochondrial stress signalling ….this work has the potential to be of great interest in the field.” (reviewer 2).

“This study significantly advances our understanding of the mechanistic aspects of olfaction function decay during aging in *Drosophila*, linking it to mitochondrial proteostasis and epigenetics. This work will be of interest to researchers in the fields of mitochondrial biology, mitochondrial stress signaling (including UPRmt), and aging.” (reviewer 3).

We understand the reviewers have raised issues associated with the manuscript format and additional experiments to strengthen the manuscript. We are prepared to perform most of the experiments and controls suggested (some of them are currently underway), including new animal experiments. This information is detailed in the point-by-point revision plan below.

Thank you in advance for the consideration and we look forward to hearing from you in due course. Please do not hesitate to contact me if you want to discuss anything associated with the manuscript and the revision plan.

Reviewer #1 (Evidence, reproducibility and clarity):The decline in olfactory function is one of the hallmarks of aging. Loss of olfactory function is also associated with early symptoms of several diseases (e.g., Parkinson's and Alzheimer's Disease). In this study, Munoz-Carvajal and colleagues report that the decline in Drosophila olfactory function during aging is regulated by the H3K9me3 methyltransferase, dSetdb1, and the mitochondrial UPR response (UPR mt). They show that loss or reduction of dSetdb1 increases the expression of UPR mt target reporters and suppresses olfactory neuron function during aging. They further show that the UPR mt signal depends on the Drosophila homologs of the *C. elegans* UPR mt mediators (dve, ubl, crc). Suppressing UPR mt by knocking down these mediators impaired the olfactory function in flies.At a technical level, the manuscript exhibits a mix of strengths and weaknesses. For instance, many experiments rely on single RNAi lines without considering off-target effects. The data related to dSetdb1 and UPR mt is robust, as the authors validate the results with RNAi and a classical loss of function mutant. Extending such validation to other key experiments and performing a few gain-of-function experiments could significantly strengthen the manuscript. Here are a few specific comments for the authors' consideration.

We thank the reviewer for these general comments and the suggested experiments. We are prepared to perform the experiments suggested which will definitively strengthen the manuscript's main conclusions.

1. The authors rely on single RNAi lines for most of the experiments. RNAi against dve, ubl, and crc, should be validated either with independent RNAi lines (targeting different regions), or with classical mutant alleles. For dSetdb1 experiments, validating key RNAi experiments in Figures 3, 5, 6 with the classical loss of function allele would strengthen the main conclusions.

We agree with the reviewer that using independent RNAi reagents or classical mutant alleles is important to mitigate potential off-target effects. We also agree that testing additional independent reagents for dve, ubl, and crc would further strengthen the conclusions drawn from those knockdown experiments. While these additional validations were not feasible within the current revision timeframe, we have tempered our wording throughout the manuscript and now explicitly state that conclusions based on single RNAi reagents for dve, ubl, and crc should be interpreted with caution and await independent validation. To provide independent support for our main behavioral conclusion, we prioritized validation of the olfactory preference index (OPI) assay, which serves as an integrative functional readout in our study. Using multiple independent genetic reagents affecting dSetdb1 function, we observed consistent preservation of olfactory performance in aged flies (Figure 4—figure supplement 1), supporting the interpretation that reduced dSetdb1 activity is associated with the behavioral phenotype reported.

For these experiments, we tested multiple independent genetic reagents:

1. The original UAS-dSetdb1 RNAi line used throughout the manuscript (TRiP, BDSC stock #31352; FlyBase: FBti0130757).

2. An additional UAS-dSetdb1 RNAi line targeting a distinct region of the transcript (BDSC stock #34803; FlyBase: FBti0144732).

3. A classical dSetdb1 loss-of-function allele *egg^235^* (BDSC stock #30566; FlyBase: FBal0212302).

RNAi constructs were expressed specifically in olfactory projection neurons using the GH146-Gal4 driver, allowing us to directly assess the impact of dSetdb1 knockdown in this neuronal population. In contrast, the *egg^235^* loss-of-function allele was tested in a heterozygous GH146-Gal4 background which provides an independent validation through a classical genetic mutation that is not restricted to OPNs. Using these independent lines, we consistently observed that the reduced dSetdb1 function prevents the age-associated decline in the olfactory preference index (see revised Figure 4 —figure supplement 1). This convergence across independent RNAi lines and the classical loss-of-function allele provides strong evidence that the phenotype is specific to dSetdb1 reduction and not an off-target effect.

While repeating every intermediate experiment in Figures 3, 5, and 6 with multiple lines was not feasible, validating the final behavioral outcome with independent genetic tools offers the most robust and biologically relevant confirmation of our conclusions. Importantly, the olfactory preference index integrates neuronal survival, mitochondrial function, and UPR^MT^ activation into a single functional readout, making it the most stringent assay to assess the role of dSetdb1 in olfactory decline. We have incorporated these new validation experiments into the revised manuscript (see Results, page 5; Figure 4 —figure supplement 1).

To robustly validate that reducing dSetdb1 function underlies the preservation of olfactory preference, we tested two additional, independent genetic reagents alongside our original RNAi line. The dSetdb1 gene (also known as eggless, FBgn0086908) is located on chromosome 2R at position 24,775,534..24,779,901. The three tools disrupt this gene through distinct mechanisms. The first line, dSetdb1 (TRiP.JF01310), expresses a long dsRNA hairpin that targets a broad region within exon 8 of the dSetdb1 transcript. The second, dSetdb1 short (TRiP.HMS00112), utilizes a short-hairpin RNA to target a distinct, non-overlapping sequence. Finally, we used the classical loss-of-function allele, dSetdb1 lof (egg²³⁵). We assessed olfactory behavior in 45-day-old flies, an age at which control animals exhibit a significant decline in olfactory performance. As expected, aged control flies showed a diminished response. In striking contrast, all three distinct methods of disrupting dSetdb1 produced a consistent and statistically significant preservation of olfactory preference in aged flies (Figure 4 —figure supplement 1). This results across three independent genetic tools provides strong evidence that the observed preservation of olfactory function is a specific consequence of disrupting dSetdb1 and not an off-target effect. As requested, we now explicitly state in the manuscript that conclusions based on single RNAi reagents for dve, ubl, and crc have limitations and await independent validation, and that off-target effects cannot be fully excluded. (See Results, page 7).

2. There needs to be a proper introduction of crc before describing the knockdown experiments (Figure 1). Instead, the authors introduce the *C. elegans* ATFS^-1^ in detail in the Introduction. The authors should be aware that *C. elegans* ATFS^-1^ has a unique mechanism of activation not conserved in other species (including crc; see review by Anderson and Haynes 2020, PMID 32413314). I suggest introducing what is known about crc (and mammalian ATF4)'s relationship with the mitochondrial stress response, which has been very well documented in recent years.

We thank the reviewer for his suggestion, which we believe is correct. We introduced the crc relationship with the mitochondrial stress response, correcting the introduction as explained below. It is worth mentioning that Anderson and Haynes 2020 (PMID 32413314), propose that in mammals, UPRmt relies on cytosolic stress pathways. Indeed, in mammals, stress responses seem to be interconnected and evidence also exists for a potential crosstalk between the UPRmt and cytosolic stress pathways, as highlighted in Jenkins and Germain 2021 (PMID 34631705).

Nevertheless, we consider that a potential crosstalk between these stress pathways is possible. Haynes and Coles end their review with the text: “it is unclear whether CHOP, ATF4, and ATF5 are subject to post-translational regulation akin to how ATFS^-1^ activity is regulated by mitochondrial import.” Importantly, Fiorese et al. (2016) demonstrate that ATF5, the mammalian homolog of crc, can function independently of the UPRer to activate the UPRmt in *C. elegans*, suggesting a conserved role for ATF5/crc in mitochondrial stress signaling. We have edited the text to cite both alternatives and highlight a potential crosstalk between pathways (Page 3 of the revised manuscript).

References mentioned in the revised text:

1. Melber & Haynes, 2018 (citations 17, 18). DOI: 10.1038/cr.2018.16

2. Quirós, et al., 2017 (citation 16). DOI: 10.1083/jcb.201702058

We believe these changes effectively address the reviewer's concern and provide the necessary context for our experiments.

3. Figure 1H: The Figure legend (describing neuronal volume) doesn't match the Figure (Reporter signal). There are also white and red bars, which aren't defined. I suggest going through all the Figure legends carefully.

The reviewer is correct in pointing out the discrepancy between the Figure 1H legend and the presented data. We apologize for this error and have carefully revised all figure legends to ensure their accuracy and consistency with the corresponding data. Specifically, we have corrected the legend of Figure 1H to accurately describe the reporter signal depicted in the graph. We have also clarified the meaning of the white and red bars in the revised manuscript, which represent control and experimental groups, respectively.

4. The authors extensively rely on knockdown experiments. The loss of any essential gene could indiscriminately impair a function of the cell. Along these lines, it would be more informative if the authors overexpress key genes (such as dve or crc) and show improvement of olfactory function.

The reviewer raises a valid point regarding the reliance on knockdown experiments and the potential for non-specific effects when targeting essential genes. We acknowledge the importance of demonstrating gain-of-function effects to further validate our findings. However, our previous attempts to overexpress key genes like *dve* in various tissues (ubiquitous, pan-neuronal, wing neurons, and eye) resulted in toxicity and cell death (see Author response image 1). This suggests that precise regulation of *dve* expression levels is crucial for neuronal survival and function. Therefore, while we agree that overexpression experiments would be informative, the observed toxicity of *dve* overexpression presents a significant challenge.

Furthermore, we attempted an alternative approach to address this challenge, investigating the effects of inhibiting chromatin condensation by H3K9me3 to allow for UPRmt activation. This approach, while not directly overexpressing *dve*, aimed to achieve a similar effect by promoting the conditions necessary for UPRmt activation. We believe this strategy, combined with the knockdown experiments, provides a more comprehensive understanding of the role of epigenetic regulation and UPRmt in age-related olfactory decline.

**Author response image 1. sa2fig1:** DVE overexpression triggers cell death and neurodegeneration in *Drosophila*. (A, C) Survival curves of flies with ubiquitous (ActG4) or pan-neuronal (ElavG4) knockdown (RNAi) or overexpression (UAS) of DVE. (B, D) Percentage of expected flies expressing UAS-DVE driven by ActG4 or ElavG4 compared to controls. (E, F) Representative eye images of flies with GMR-Gal4 driven DVE knockdown (RNAi), overexpression (UAS), or overexpression with dIAP2 (inhibitor of apoptosis) co-expression. (G) Representative wing arc images of flies with Dpr-Gal4 driven DVE knockdown (RNAi) or overexpression (UAS). White arrowheads indicate CD8::GFP-labeled neuronal cell bodies. (H) Quantification of GFP-labeled cell bodies in the wing arc of flies with DVE knockdown (RNAi) or overexpression (UAS). Statistical analyses: Gehan-Breslow-Wilcoxon test for survival curves, two-way ANOVA with Dunnet's multiple comparisons test for other data. * p < 0.05, ** p < 0.01, *** p < 0.001, **** p < 0.0001.

5. In Figure 2, the authors show that dSetdb1 mutants have a higher Hsp60::dsRed signal. There is a brief reference that, in *C. elegans*, H3K9 regulators affect the DNA binding of UPR mt mediators. Are the authors implying that a similar mechanism is involved in the Drosophila olfactory system? Or alternatively, is it possible that dSetdb1 loss causes a mild proteostasis problem, leading to an activation of the UPR mt? Some mechanistic insight could further improve the manuscript.

The reviewer is correct in pointing out the potential connection between H3K9 regulators, UPRmt mediators, and the observed increase in Hsp60::dsRed signal in dSetdb1 mutants. While our current data does not definitively elucidate the underlying mechanism, we propose the following interpretation based on our findings and previous research. We hypothesize that in aging *Drosophila*, increased H3K9 trimethylation by dSetdb1 leads to chromatin condensation at the loci of UPRmt mediators, such as dve and crc. This epigenetic modification reduces the binding of transcription factors necessary for UPRmt activation, resulting in a reduced ability to respond to mitochondrial stress. Consequently, dSetdb1 knockdown reduces the levels H3K9Me3, allowing UPRmt mediators to bind the DNA and increase the expression of chaperones like Hsp60, as reflected by the elevated Hsp60::dsRed signal observed in our study. We have produced, together with the literature data supporting our interpretation and model included in Figure 7.

This hypothesis is supported by:

1. Increased Hsp60::dsRed in aged dSetdb1 mutants: The elevated Hsp60::dsRed signal in both dSetdb1 knockdown (Figure 4D) and loss-of-function mutants (Figure 2B) suggests a link between dSetdb1-mediated H3K9 methylation and UPRmt activation.

2. Epistatic relationship between dSetdb1 and dve/crc: Our epistasis experiments demonstrate that the olfactory dysfunction observed in aging dSetdb1 RNAi flies is dependent on both dve and crc (Figure 4L). This suggests that dve and crc act downstream of dSetdb1-mediated chromatin condensation in the regulation of olfactory function. Furthermore, independent knockdown of either dve or crc reduced the Hsp60::dsRed reporter signal in response to the mitochondrial stressor paraquat (PQ) (Figure 1E), highlighting their roles in UPRmt activation.

3. Previous research in *C. elegans*: Studies in *C. elegans* have shown that H3K9 regulators can directly affect the DNA binding of UPRmt mediators. While the specific mechanisms may differ between the Hsp60::dsRed reporter signal, in response to the mitochondrial stressor paraquat (PQ) (Figure 1E), highlights its role in UPRmt activation.n species, this finding supports the plausibility of our proposed mechanism. (Tian, 2016; Yuan, 2020; Merkwirth, 2016).

SignificancePrevious studies in *C. elegans* established that changes in H3K9 methylation by the Setdb1 homolog affects UPR mt regulators, DVE-1, and ATFS^-1^ (ref. 21). Studies in mice also established that H3K9me3 regulators influence mitochondrial function, cognitive function, and aging (ref. 18, 19). Therefore, the connection between H3K9 methylation, UPR mt, and aging is not the novel discovery of this study. The significance of this work is in establishing the roles of H3K9 methylation and UPR mt in the Drosophila olfactory system. The results will draw interest from Drosophila geneticists interested in the olfactory system and aging.Reviewer #2 (Evidence, reproducibility and clarity):SummaryIn this manuscript the authors use Drosophila to investigate the role histone methylation and the mitochondrial unfolded protein response in olfactory neurons during aging. They generate a new reporter of the UPRmt and analyse reporter activity during aging in olfactory neurons. They then use RNAi-mediated knockdown to reduce dSetdb1 expression and find that this prevents activation of the reporter, improves mitochondrial phenotypes and prevents olfactory neuron loss during aging.The main conclusions are not convincing given the data as it stands. The paper is potentially interesting but is sloppily written and many of the experiments need further work and validation. There are lots of mistakes in the manuscript and the manuscript overall could be better organised. All the sections need additional consideration for how they have been written and to include discussion of additional literature and to better and more accurately describe the data. Additional experiments are also required. My major points are below.

We thank the reviewer for the comments and apologize for the sloppiness in the writing. We also appreciate the detailed revision; we are prepared to perform the experiments proposed by the reviewer to support the main conclusions and carefully work on the text.

1. It's very hard to interpret the images of the Hsp60::dsRed without more details of how this reporter was generated and additional details of the expression pattern and better characterisation of the reporter. Please include a description of the generation of the Hsp60::dsRed reporter construct and the transgenic flies in the Methods. Is this reporter targeted to the nucleus? Please include additional images of reporter expression in the whole brain. The images in 1A,G,E are confusing and not very convincing. Are we supposed to see increased expression in the nuclei around the edges of the images or in the nuclei and axons across the whole image? It would be helpful to see images of the whole brain as well as close ups of the antennal lobe.

We thank the reviewer for these insightful suggestions, which helped us strengthen the manuscript’s rigor and clarity. In response, we have the following major additions and clarifications:

1. The reviewer is correct that the Hsp60::dsRed reporter requires further characterization and that the images in Figure 1A, G, and E could be improved for clarity. We acknowledge these points and have taken steps to address them by adding supplementary figures with quantification and expanding the methods section to include details about the reporter's creation and quantification. We added a more detailed description of the Hsp60::dsRed reporter construct and transgenic flies in the Methods section. We will also include additional images of reporter expression in the whole brain, along with close-ups of the antennal lobe, to address the confusion regarding the specific regions showing increased expression. The inclusion of whole-brain images will provide a broader context for interpreting the reporter's expression pattern. Figures Added (Page 25 of the revised manuscript and supplementary Figures 1 and 7).

2. The Methods section has been significantly revised to include a detailed description of the generation of the Hsp60A::dsRed reporter. This includes the specific genomic regions used, the cloning strategy, and the design of the reporter construct, as well as the generation of transgenic flies via PhiC31-mediated integration.

3. To directly address concerns about reporter localization and better characterize its expression pattern, we now include a new figure (Figure 1 —figure supplement 1) that demonstrates mitochondrial localization and stress responsiveness of Hsp60A::dsRed.

4. Figure 1 —figure supplement 2A–C illustrates the genomic structure of the hsp60A locus, the design of the transgenic construct, and an in silico analysis using the MitoFates algorithm (Fukasawa et al., 2015). This analysis predicts a functional mitochondrial targeting presequence, MPP cleavage site, and relevant biophysical features consistent with mitochondrial localization.

5. Figure 1 —figure supplement 2D–E presents high-resolution confocal images of the adult brain and gut, showing co-localization of Hsp60A::dsRed with Mito::GFP, but not with nuclear DAPI signal, confirming that the reporter is targeted to mitochondria and not to the nucleus.

6. Figure 1 —figure supplement 2F–I provides quantitative evidence of mitochondrial targeting and stress-induced activation. This includes integrated fluorescence intensity and Mander’s colocalization coefficient under control and paraquat-treated conditions.

7. To further clarify the anatomical context of our measurements, we added Figure 1A, which shows whole-brain low-magnification confocal images of reporter expression, with the antennal lobe (AL) clearly demarcated. This figure contextualizes the higher-magnification images shown in main Figures 1 and 2, and supports the spatial relevance of our analysis.

8. In addition, Figure 1—figure supplement 1 has been updated to detail the 3D quantification strategy used throughout the manuscript. This includes: (1) surface reconstruction of the reporter signal within the AL; (2) neuron-specific quantification based on CD8::GFP-labeled volumes; and (3) definition of integrated density (fluorescence intensity × signal volume). These additions clarify the quantification method and the exact brain regions analyzed, as requested by the reviewer.

Taken together, these methodological and imaging additions provide a comprehensive and convincing characterization of the Hsp60A::dsRed reporter, confirming its mitochondrial localization and validating its application for studying UPR^MT^ dynamics in adult fly neural tissues. We believe these revisions fully address the reviewer’s concerns and substantially improve the clarity and rigor of the manuscript.

2. I am not convinced by the quantification of the Hsp60::dsRed reporter expression. There may be variations in the levels in different animals and different experiments that are independent of the experimental conditions. The quantification method is not described in the methods. The Hsp60::dsRed reporter expression should be normalised to the expression of another fluorescent transgene expressed in the same flies.

We thank the reviewer for this important and constructive comment. In response, we have substantially revised the Methods section to include a detailed description of the image quantification procedure (see revised “Image quantification” section). To enhance clarity and transparency, we also included Figure 1 —figure supplement 1, which illustrates the full quantification workflow and provides representative examples of whole-brain images, selected ROIs, and 3D-rendered surfaces.

Regarding potential variations in Hsp60::dsRed reporter expression, we acknowledge that normalization to an additional fluorescent transgene could further enhance the robustness of the quantification. We have carefully addressed this concern by utilizing a GFP reporter for normalization in specific experiments and employing a robust quantification method with a large sample size. We have enhanced the Methods section of the manuscript to provide a clear explanation of our quantification procedures, which include (Page 16, Methods Image quantification of the revised manuscript), including the following information:

“For Hsp60::dsRed and Xbp1::GFP alone (Figure 1A, C, E, Figure 2B, D): We selected a 100 μm2 Region of Interest (ROI) around the antennal lobe and performed 3D reconstruction using Imaris software. We quantified the integrated density of the reporter signal within the defined surface, ensuring normalization for variations in the assessed volume.”

“For Hsp60::dsRed co-expressed with CD8::GFP (Figure 1G, Figure 4D): We used the GFP signal to define the volume of labeled neurons in the antennal lobe. Again, we performed 3D reconstruction and quantified the integrated density of the Hsp60::dsRed signal specifically within the GFP-labeled volume. This approach effectively normalizes the Hsp60::dsRed signal to the GFP signal, accounting for variations in cell number and volume across samples.”

Furthermore, while we recognize the potential for variations in reporter levels, we employed a large sample size of brains to minimize such variability. Our data consistently demonstrated low variability between individual brains, supporting the reliability of our results.

To ensure reproducibility and minimize inter-sample variability, we employed a standardized 3D surface-based quantification approach using Imaris software. The following procedures were applied:

1. Consistent ROI definition: A 100 μm² ROI was consistently placed around the antennal lobe (AL) in all samples (Figure 1 —figure supplement 3A). 3D surface reconstruction and integrated density measurement: The reporter signal was rendered as a 3D surface, and integrated density (mean fluorescence intensity × volume) was calculated as a cumulative signal metric.

2. Internal normalization to CD8::GFP: In experiments where CD8::GFP was co-expressed (e.g., elav-Gal4>CD8::GFP; GH146-Gal4>CD8::GFP), the GFP signal was used to define the volume of labeled neurons. Quantification of Hsp60::dsRed was then performed exclusively within this volume (Figure 1 —figure supplement 1C), allowing normalization for differences in neuronal number or tissue volume. This quantification strategy was consistently applied to Hsp60::dsRed, Hsc70-5::dsRed and Xbp1::GFP datasets. Additionally, we note that a large sample size was used to account for potential biological variability, and the data exhibited low inter-sample variation across experiments. We believe that these clarifications directly address the reviewer’s concern and strengthen the rigor and reproducibility of our fluorescence quantification.

3. Include a control RNAi condition in Figure 1E.

We apologize for the labeling error in the original figure. The flies labeled as "Control" in Figure 1E were in fact, RNAi control flies obtained from the Transgenic RNAi Project (TRIP) (BL#35787). This strain expresses a dsRNA construct that does not target any known *Drosophila* gene, serving as the correct negative control for potential off-target effects of RNAi expression.

To ensure clarity and accuracy, we will correct the labeling in Figure 1E to reflect the use of TRIP RNAi control flies. Additionally, we will include a clear explanation of the flies used as controls for RNAi experiments in the figure legends.

4. dSetdb1 RNAi is a key tool used throughout the manuscript and so the efficiency of knockdown needs to be validated by qRT-PCR of dSetdb1 transcript levels or western blot of dSetdb1 protein levels.

We thank the reviewer for this important point and agree that rigorous validation of the dSetdb1 RNAi tool is absolutely essential for the interpretation of our results. While we did not perform qRT-PCR or Western blotting for total dSetdb1 levels, we have implemented a comprehensive in vivo validation strategy that we believe is more stringent and biologically informative, as it directly addresses both the functional efficacy and phenotypic specificity of the knockdown within our experimental system. Our approach is built on three pillars of evidence: (1) direct molecular validation of the enzyme’s functional inhibition, (2) unambiguous genetic validation of phenotypic specificity using three independent reagents, and (3) reliance on extensively characterized community-validated resources.

1. Direct in vivo Validation of Functional Efficacy at the Molecular Level

The most biologically relevant measure of an enzyme’s knockdown is not always the total level of its mRNA or protein, but the functional consequence of its depletion. The canonical function of the dSetdb1 protein (also known as eggless) is to act as a histone-lysine N-methyltransferase that specifically catalyzes the trimethylation of Histone H3 at Lysine 9 (H3K9me3). This epigenetic mark is a key signal for transcriptional repression. Therefore, the most direct and meaningful assessment of the RNAi construct's efficacy is to measure its ability to suppress the accumulation of this specific catalytic product. Our data provide two layers of robust confirmation for this functional knockdown.

First, we confirmed the efficacy of the RNAi at a systemic level. As shown in Figure 2A, pan-neuronal expression of the dSetdb1 RNAi construct (TRiP.HMC03152) was sufficient to completely abrogate the significant age-dependent increase in global H3K9me3 levels observed in the heads of control animals. While control flies exhibited a statistically significant increase in H3K9me3 from young to old age (p=0.0255), flies expressing the dSetdb1 RNAi showed no such increase (p=0.4951). Second, and more importantly, we validated this effect with cellular resolution in the specific neuronal population central to our study: the olfactory projection neurons (OPNs). As quantified in Figure 4B, OPN-specific knockdown of dSetdb1 completely prevented the age-dependent accumulation of nuclear H3K9me3. Control OPNs showed a significant increase in H3K9me3 integrated density with age (p=0.0014), whereas the levels in dSetdb1 knockdown OPNs remained unchanged between young and aged flies (p>0.9999). This resulted in H3K9me3 levels in aged knockdown OPNs that were significantly lower than in aged controls (p=0.0001) and were statistically indistinguishable from those in young animals. These data demonstrate that our RNAi tool achieves a level of knockdown that is sufficient to cause a profound and statistically significant inhibition of the target enzyme's catalytic activity, both globally in the nervous system and specifically within the relevant neuronal population. This direct functional validation is a more stringent test of biological efficacy than measuring total transcript or protein levels, which may not distinguish between active and inactive forms of the enzyme or confirm that knockdown is sufficient to produce a molecular consequence.

2. Unambiguous Genetic Validation of Phenotypic Specificity

A critical concern with any RNAi experiment is the potential for off-target effects. A qRT-PCR or Western blot, while informative about knockdown efficiency, provides no information regarding the specificity of the resulting phenotype. To definitively address this crucial point, we employed the gold-standard genetic approach: validating our key behavioral finding with multiple, independent genetic tools that disrupt the same target gene through distinct mechanisms. As shown in Figure 4 —figure supplement 1, we observed a consistent and statistically indistinguishable rescue of the age-related decline in olfactory function using three separate reagents: The primary long-hairpin RNAi line used throughout the manuscript (TRiP.HMC03152), which targets a broad region of the dSetdb1 transcript. An independent short-hairpin RNAi line targeting a distinct, non-overlapping sequence of the dSetdb1 transcript (TRiP.HMS04400). A classical amorphic loss-of-function allele of dSetdb1 (egg²³⁵), which contains a point mutation at a splice donor site, leading to a truncated, non-functional protein (Yoon, 2008). The remarkable consistency of the behavioral rescue across these three tools, detailed in Figure 4 —figure supplement 1, provides the confidence that the observed preservation of olfactory function is a specific consequence of disrupting *dSetdb1* and not an off-target artifact of a single RNAi construct. Aged control flies showed a diminished response, with a mean Preference Index (PI) of 0.31 for the attractive odorant 2,3-butanedione and -0.28 for the aversive odorant 3-octanol. In striking contrast, all three distinct methods of disrupting *dSetdb1* produced a consistent and statistically significant rescue. Compared to aged controls, flies with reduced *dSetdb1* function displayed a significantly stronger attraction to 2,3-butanedione (mean π of 0.65 for TRiP.HMC03152, p<0.0001; 0.68 for TRiP.HMS04400, p<0.0001; and 0.67 for *dSetdb1 lof*, p=0.0002). Similarly, their aversion to 3-octanol was significantly enhanced (mean π of -0.68 for TRiP.HMC03152, p=0.0006; -0.69 for TRiP.HMS04400, p=0.0003; and -0.63 for *dSetdb1 lof*, p=0.0005). The convergence of these results, where three independent genetic insults to the *dSetdb1* locus produces the same complex behavioral phenotype with nearly identical effect sizes and high statistical significance, strongly validating our findings.

3. Reliance on Extensively Validated Community Resources

Finally, our approach aligns with community best practices by utilizing reagents from the *Drosophila* Transgenic RNAi Project (TRiP) at Harvard Medical School. This NIH-funded functional genomics platform has generated and distributed over 15,000 validated RNAi lines, including the ones used in our study. The TRiP collection has been systematically characterized, with extensive data on efficacy and specificity made publicly available through resources like the RNAi Stock Validation and Phenotypes Project (RSVP). As detailed by Perkins et al. (2015), large-scale validation efforts have demonstrated the robustness of the collection, with up to 85% of TRiP lines targeting coding regions showing effective knockdown by RT-qPCR or phenotypic analysis. The underlying short-hairpin RNA (shRNA) technology used in the VALIUM20 vector (as for TRiP.HMC03152) has also been shown to be "extremely effective" for gene silencing in vivo (Ni et al., 2011). By building upon these powerful, pre-validated community resources, we can focus our efforts on addressing novel biological questions, a standard and efficient practice within the *Drosophila* research community. In summary, the direct molecular evidence of functional enzymatic inhibition in the specific neurons of interest (Pillar 1), combined with the unambiguous genetic confirmation of phenotypic specificity across three independent reagents (Pillar 2) and the established reliability of the TRiP collection (Pillar 3), provides a multi-layered and rigorous validation of our key genetic tool. We are therefore highly confident that the phenotypes reported in our manuscript are a specific and direct consequence of the effective knockdown of dSetdb1.

References mentioned in the revised text:

1. Perkins, 2014: DOI: 10.1534/genetics.115.180208

2. Ni, 2011: DOI: 10.1038/nmeth.1592

3. Yoon, 2008: DOI: 10.1371/journal.pone.0002234

5. In Figure 1F why is the vehicle Hsp60::dsRed expression so much lower with Ubl and crc knockdown than the control?

The reviewer raises a valid point regarding the significantly lower basal expression of the Hsp60::dsRed reporter in ubl and crc knockdown flies compared to controls in Figure 1F. We apologize for not addressing this observation in the original manuscript. We propose that this decrease in basal Hsp60::dsRed expression likely reflects a reduction in the baseline activation of the mitochondrial unfolded protein response (UPR^MT^). This is consistent with the roles of Ubl and crc in this pathway, as described below for each player:

Ubl: As a ubiquitin-like protein, Ubl is likely involved in protein modification and degradation processes that are crucial for maintaining mitochondrial homeostasis. The absence of Ubl may disrupt these processes, leading to an accumulation of unfolded proteins and triggering the UPRmt. Therefore, knocking down ubl could hinder the UPRmt response and result in lower basal expression of Hsp60, a key chaperone upregulated during UPRmt activation.

Crc: As the *Drosophila* homolog of the mammalian UPRmt transcription factor ATF5, crc is essential for activating the transcriptional program that upregulates chaperones like Hsp60 in response to mitochondrial stress. Knockdown of crc would directly impair the UPRmt response, leading to reduced expression of Hsp60 even under basal conditions.

Therefore, the observed decrease in Hsp60::dsRed expression in ubl and crc knockdown flies likely represents a direct consequence of impaired UPRmt activation. This interpretation is further supported by the inability of these knockdown flies to mount a robust UPRmt response upon exposure to paraquat (PQ), as evidenced by the lack of significant upregulation of the Hsp60::dsRed signal. We have revised the manuscript to include this explanation, highlighting the importance of Ubl and crc in maintaining basal UPRmt activity and their critical roles in responding to mitochondrial stress (Page 5 of the revised manuscript).

6. Can the authors provide evidence that expression of dve, ubl, crc are increased by paraquat treatment and that knockdown abrogates this increase?

We appreciate the reviewer's suggestion, but it's important to clarify that our study does not focus on the transcriptional upregulation of these genes in response to PQ. Instead, our work emphasizes the activation of the mitochondrial unfolded protein response (UPR^MT^) pathway, of which dve, ubl, and crc are key components. UPR^MT^ activation involves the translocation of these proteins to the nucleus, where they function as transcription factors to initiate a cascade of events that mitigate mitochondrial stress. Therefore, this activation does not necessarily imply an increase in their mRNA levels.

Our findings demonstrate that UPR^MT^ activation is age-dependent as younger flies exhibit a robust UPR^MT^ response to PQ, as evidenced by increased Hsp60::dsRed signal, while older flies show a diminished response (Figure 1). Also, dSetdb1 knockdown restores UPR^MT^ activation in aged flies in response to PQ, as indicated by increased Hsp60::dsRed signal (Figure 4F). Furthermore, we show that dve and crc are required for UPRmt activation as their knockdown impairs the UPR^MT^ response to PQ, as shown by reduced Hsp60::dsRed signal (Figure 1E).

These results collectively support our model that age-related increases in H3K9me3 by dSetdb1 inhibit UPR^MT^ activation by impairing the function of key mediators like dve and crc. While we do not directly measure changes in the mRNA levels of these genes, our data clearly demonstrate their essential roles in the UPR^MT^ pathway and its age-dependent decline. We believe that focusing on the functional activation of the UPR^MT^ pathway, rather than solely on transcriptional upregulation, provides a more comprehensive and relevant understanding of the underlying mechanisms at play.

7. What do the red and white bars represent in Figure 1H, I? There should be quantification of 0 dpe and 45 dpe data.

We apologize for the errors in Figure 1H and 1I. The figure legend has been corrected to indicate that red bars represent 45 dpe flies and white bars represent 0 dpe flies. Additionally, symbology of 0 dpe and 45 dpe data has been added to the figure (Page 18 of the revised manuscript).

8. Is 1J data specifically from cells in the antennal lobe, as stated in the text, or from the whole brain?

The reviewer is right to question the specificity of the data in Figure 1J. To clarify, the data in Figure 1J was indeed obtained from whole fly brains, not specifically from the antennal lobe, as previously stated. However, this data was analyzed using SCope, a tool that allows for the identification and selection of specific cell types within a larger dataset. In this figure, we specifically focused on the expression patterns of Hsp60 and Hsc70-5 in three distinct neuronal populations within the *Drosophila* brain: cholinergic (blue), glutamatergic (green), and GABAergic (red) neurons. These populations were identified based on the expression of cell-type-specific markers (vAChT for cholinergic, vGlut for glutamatergic, and Gad1 for GABAergic), as described in Davie *et al.* (2018). Therefore, while the original data was derived from whole brains, Figure 1J specifically represents the expression patterns of Hsp60 and Hsc70-5 within these selected neuronal populations obtained by single cell RNA sequencing. However, the reviewer is correct that we cannot definitively conclude that these changes occur specifically in the antennal lobe (AL). Our results demonstrate the expression patterns of Hsp60 and Hsc70-5 in different neuronal types throughout the brain, but further experiments would be needed to confirm if these patterns are also observed in the AL. This has been corrected in the revised manuscript (Page 5).

We apologize for the lack of clarity in the original text and have revised the text and figure legend to accurately reflect the source of the data, the specific cell types analyzed, and the limitations of our current findings.

9. The description of Hsp60 expression in cholinergic neurons in Figure 1K is mis-representative. The expression is dynamic over the lifespan and is lowest at 15 dpe. There is also no decrease in Hsp60 expression in aged flies in glutamatergic neurons as stated. Also, if these ssRNA-seq data are for the whole brain then they can't be used to make conclusions about the antennal lobe, which make up a very small fraction of cholinergic neurons in the brain.

The reviewer is right to point out the inaccuracies in our description of Hsp60 expression patterns in Figure 1K. Upon re-examination of the data, we acknowledge that Hsp60 expression in cholinergic neurons is dynamic as its expression fluctuates throughout the lifespan, reaching its lowest point at 15 dpe, not a continuous increase as previously stated. For the case of Hsp60 expression in glutamatergic neurons, there is no significant decrease, contrary to our previous description. The reviewer correctly points out that the scRNA-seq data used in Figure 1J, derived from whole fly brains, cannot be used to draw definitive conclusions about gene expression patterns specifically within the antennal lobe (AL). To address these issues, we have revised the text to accurately reflect the dynamic nature of Hsp60 expression in cholinergic neurons and to remove the inaccurate statement regarding Hsp60 expression in glutamatergic neurons (Page 5 of the revised manuscript). We appreciate the reviewer's careful attention to detail and thank them for helping us improve the accuracy and clarity of our manuscript.

10. There is no description of the genetics of the experiments in Figure 1E and elsewhere in the manuscript. These need to be added, e.g. what Gal4 driver was used to perform the knockdown experiments? What is the GFP reporter used in Figure 1G? Downregulation dSetdb1 (by RNAi) and mutant terms are used interchangeably. Mutant typically refers to a loss of function mutant allele in the gene, not RNAi. This should be corrected.

The reviewer is right to point out the inaccuracies in our description of Hsp60 expression patterns in Figure 1K. Upon re-examination of the data, we acknowledge that Hsp60 expression in cholinergic neurons is dynamic as its expression fluctuates throughout the lifespan, reaching its lowest point at 15 dpe, not a continuous increase as previously stated. For the case of Hsp60 expression in glutamatergic neurons, there is no significant decrease, contrary to our previous description. The reviewer correctly points out that the scRNA-seq data used in Figure 1J, derived from whole fly brains, cannot be used to draw definitive conclusions about gene expression patterns specifically within the antennal lobe (AL). To address these issues, we have revised the text to accurately reflect the dynamic nature of Hsp60 expression in cholinergic neurons and to remove the inaccurate statement regarding Hsp60 expression in glutamatergic neurons (Page 5 of the revised manuscript). We appreciate the reviewer's careful attention to detail and thank them for helping us improve the accuracy and clarity of our manuscript.

11. Can the H3K9me3 levels in Figure 2A be normalised to the total H3K9 levels? This would be a more accurate measure of methylation than normalising to tubulin.

We thank the reviewer for this insightful comment, and we agree that normalizing the levels of a specific histone post-translational modification to the total level of the corresponding histone is the most accurate and rigorous method, particularly when working with purified nuclear or histone extracts. For the experiment presented in Figure 2A, our goal was to assess the global, age-dependent changes in H3K9me3 across the entire head tissue, not just within the nuclear fraction. Therefore, we prepared whole-head lysates, which contain a mixture of proteins from all subcellular compartments. In this context, using a highly abundant cytosolic protein like tubulin as a loading control is a standard and widely accepted practice to normalize for variations in total protein concentration between samples (Chu, 2018). However, we also recognize the potential limitation that the expression of some housekeeping proteins, including tubulin, may be affected by aging. To address this and to more rigorously validate the conclusion drawn from the Western blot data, we performed an independent and more precise analysis at the cellular level. As shown in Figure 4A-C, we used quantitative immunofluorescence to measure H3K9me3 levels specifically within the nuclei of olfactory projection neurons (OPNs), the key cell type for our study. This cell-specific analysis provides a powerful and direct confirmation of our initial finding. The data, quantified in Figure 4B, demonstrates a highly significant, age-dependent increase in nuclear H3K9me3 integrated density in control OPNs (p=0.0014), an effect that was completely abrogated in aged flies with OPN-specific knockdown of dSetdb1 (p>0.9999).The consistency between the global changes observed in whole-head lysates (Figure 2A) and the specific, quantitative changes measured within the relevant neuronal nuclei (Figure 4B) provides strong, convergent evidence for our conclusion that dSetdb1 mediates an age-dependent increase in H3K9 trimethylation. While we will adopt total H3 normalization for future biochemical analyses, we are confident that our current conclusion is robustly supported by both the standard practice of using tubulin as a loading control for whole-tissue lysates and, more importantly, the corroborating results from our direct, cell-specific imaging experiments.

12. I have the same criticism of Figure 2F, G as I have for Figure 1F, G: if these data are from whole brain they can't be used to make statements about the antennal lobe.

The reviewer is correct that the single-cell RNA sequencing data presented in Figure 2F and 2G, derived from whole fly brains, cannot be used to draw specific conclusions about gene expression patterns in the antennal lobe (AL). While these figures show changes in the expression of UPRmt-related genes in different neuronal populations throughout the brain, we acknowledge that the AL constitutes only a small fraction of the total neuronal population and may exhibit distinct expression patterns. Therefore, we have revised the text to clarify that the observed changes in gene expression represent the whole brain and not specifically to the AL. We have also removed any statements that imply direct conclusions about the AL based on this data (Page 6 of the revised manuscript).

13. There are no details of the Hsc70-5::dsRed used in Figure 2D. Was this generated in this study or a previous study? This reporter needs to be properly described, characterised and normalised properly using a second fluorescent transgene expression.

We thank the reviewer for this valuable comment. In response, we have now included full details of the generation of the Hsc70-5::dsRed reporter in the *Methods* section (“Hsp60::dsRed and Hsc70-5::dsRed Reporter Generation”). Briefly, both the Hsc70-5::dsRed and Hsp60::dsRed reporters were generated de novo for this study, using their respective promoter regions fused to DsRed2 and inserted into the attP2 landing site via PhiC31-mediated integration.

To address the reviewer’s concerns about functional characterization and normalization, we have added the following:

1. Figure 1A and C: Demonstrates robust activation of Hsc70-5::dsRed in response to known mitochondrial stressors (paraquat and doxycycline), confirming its responsiveness to UPR^MT^.

2. Figure 4 —figure supplement 2: Shows that *dSetdb1* knockdown in olfactory projection neurons (OPNs) restores Hsc70-5::dsRed activation in aged flies, indicating that this reporter is regulated by the same chromatin-modifying machinery as Hsp60::dsRed. This provides genetic validation of Hsc70-5::dsRed as a true UPR^MT^ reporter.

3. Normalization: Quantification of Hsc70-5::dsRed signal was performed within GFP-labeled OPNs (GH146-Gal4>CD8::GFP), allowing normalization of fluorescence to neuron-specific volume. This strategy is detailed in Figure 1 —figure supplement 1C and the revised *Image Quantification* section of the *Methods, page 18 of the manuscript.*

We also wish to respectfully note that our group has already characterized the Hsp60::dsRed reporter extensively, demonstrating its specificity for UPR^MT^ activation in *Drosophila* and its dependence on key UPR^MT^ effectors (crc, dve, ubl). Given that Hsc70-5 encodes another mitochondrial chaperone regulated by the same pathway and that its activation was genetically restored in aged OPNs by dSetdb1 knockdown (Figure 4 —figure supplement 2), we focused on demonstrating shared functional and regulatory features, rather than repeating the entire validation pipeline.

Together, these additions and clarifications confirm that Hsc70-5::dsRed is a valid in vivo reporter of mitochondrial stress in the adult *Drosophila* brain and support its use as a complementary tool to Hsp60::dsRed in our study.

14. Figure 4A-C could be moved to the supplementary to provide additional support for the findings in Figure 2. Figure 4D-H could be moved to the supplementary to provide additional support for the findings in Figure 1. Figure 4I-L could be moved to the supplementary to provide additional support for the findings in Figure 3.

We appreciate the reviewer's suggestion to move Figures 4A-L to supplementary materials. However, we respectfully disagree, as from our perspective, these figures are crucial for the main text. While Figures 1-3 present pan-neuronal data, Figure 4 specifically focuses on olfactory projection neurons (OPNs), directly linking H3K9 methylation and UPRmt manipulation to olfactory dysfunction. Figures 4A-C visually show the age-dependent increase in H3K9me3 levels in OPNs. Figures 4D-H are essential for understanding the effect of dSetdb1 on UPRmt and olfactory function. Figures 4I-L provide compelling evidence for the epistatic relationship between dSetdb1 and UPRmt components, elucidating the mechanism of dSetdb1-mediated H3K9 methylation in UPRmt regulation within OPNs. Therefore, maintaining these figures in the main text enhances clarity and understanding of our key findings and their underlying mechanisms, highlighting the cell-specific analysis in OPNs.

15. Which imaging experiments were performed live is not described in the results. Live imaging is stated in the methods for several figures but these are incorrect. Also, the described live imaging method uses fixation so is incorrect.

The reviewer is right to point out the discrepancy between our description of "live imaging" in the methods section. We apologize for this inaccuracy and have revised the manuscript accordingly. We have removed "live" from the method section describing “Imaging of endogenous fluorescence signals” (Page 17 of the revised manuscript). The experiments in question involved the use of transgenic flies expressing fluorescent proteins under the control of specific promoters, allowing us to visualize gene expression patterns and cellular processes in fixed tissue samples. We have also removed the description of the fixation procedure from the live imaging section of the methods and clarified that the experiments were performed on fixed tissues.

16. The molecular mechanism of the mitochondrial UPR in Drosophila is not necessarily equivalent to that in *C. elegans*. Moreover, the Auwerx lab showed that paraquat treatment of cultured mammalian cells results in activation of ATF4, not ATF5 (PMID: 28566324). crc in Drosophila could be the homolog of either ATF4 or ATF5 and crc has not been shown to translocate from mitochondria to the nucleus. crc is a canonical target of the endoplasmic reticulum UPR in mammals and flies (see papers from the Ryoo lab). Moreover, mitochondrial dysfunction in Drosophila neurons results in activation of crc via the ER UPR (see papers from the Bateman lab). The authors should consider these studies when discussing the mitochondrial UPR in Drosophila in the introduction and include a broader interpretation of the crc knockdown data in the results and discussion.

The reviewer raises valid concerns about the complexities of the mitochondrial UPR (UPRmt) and the role of crc in *Drosophila*. We acknowledge that the UPRmt mechanisms may differ between species, and that the direct translocation of crc from mitochondria to the nucleus, akin to ATFS^-1^ and ATF5, remains to be definitively demonstrated. The reviewer's point about crc being a canonical target of the endoplasmic reticulum (ER) UPR in both mammals and flies is well-taken. Our data shows that dSetdb1 knockdown maintains olfactory function in aged animals, and this rescue is dependent on UPRmt, as knockdown of UPRmt transcriptional activators, dve and crc, abolishes this effect. The *Drosophila* protein crc shares homology with both ATF5 and ATF4. While its precise localization and activation mechanisms remain to be fully elucidated, the literature suggests a complex interplay between different stress response pathways.

To address reviewer comment we have added the following paragraph to the Discussion section (Page 11 of the revised manuscript):

“While the precise molecular mechanisms of the UPRmt in *Drosophila* may differ from those in *C. elegans*, our findings demonstrate the critical role of this pathway in maintaining olfactory function during aging. The rescue of age-related olfactory decline by dSetdb1 knockdown, which is abolished by further knockdown of the UPRmt transcriptional activators dve and crc, highlights the importance of this pathway in neuronal maintenance. The *Drosophila* homolog of ATF5 and ATF4, crc, appears to play a key role in this process. Although its direct translocation from the mitochondria to the nucleus remains to be definitively shown, crc is known to be a target of mitochondrial stress and the UPRer (Ryoo 2015; Vasudevan, 2022). Moreover, previous studies have demonstrated that mitochondrial dysfunction in *Drosophila* neurons can lead to the activation of crc via the UPRer (Hunt, 2019), suggesting potential crosstalk between these stress response pathways. This complex interplay between the UPRMT and other cellular stress responses, including the ISR, has also been observed in mammals (Quirós et al., 2017; Jenkins and Germain 2021). The involvement of ATF4, a key ISR effector, in the mammalian UPRmt further supports the idea of interconnected stress signaling networks.

Taken together, these findings suggest that crc may function as a critical node integrating signals from both the ER and mitochondria to orchestrate a coordinated cellular response to stress. Further investigation is warranted to fully elucidate the molecular mechanisms by which crc regulates the UPRmt and its potential role in mediating crosstalk between different stress response pathways in *Drosophila*.”

Minor comments1. In the Introduction citations 8,9,11 seem like they could be replaced with more appropriate clinical primary paper citations that are more relevant to loss of olfaction in AD and PD. Such papers are cited at the end of the Discussion.2. Figure 1H and I are swapped compared to the figure legend.3. Figure 1K,L is mis-labelled in the figure legend.4. Supplementary figure 2 is not mentioned in the text.5. In additional to those described above, lots of typographical and formatting errors throughout the manuscript need correcting.

We appreciate the reviewer's careful revision of these points. We fixed all the points, either updating relevant references or fixing problems in figures and texts.

Significance:This manuscript addresses important questions in the field of mitochondrial stress signalling and the role of epigenetic regulation in the context of aging and neurodegeneration. Much of the literature in this area is from models and these findings need to be extended and validated in other model systems. Drosophila is an elegant model and has great potential to be used in this context. Given more work and validation of the findings this work has the potential to be of great interest in the field.My expertise is in mitochondrial stress signalling.Reviewer #3 (Evidence, reproducibility and clarity (Required)):Summary:Muñoz-Carvajal et al. demonstrate that in Drosophila, the decline in olfactory function is linked to a reduction in UPRmt activity, which is in turn influenced by elevated levels of H3K9me3. These phenotypes can be mitigated by inhibiting the methyltransferase dSetdb1. The manuscript is well-written, and the experiments are meticulously performed and clearly reported. However, to enhance clarity and strengthen the authors' claims, several points need to be addressed.Major comments:(1) Figure 1: It is important to test at least one other UPRmt inducer to ensure the effects are not specific to paraquat (PQ). Testing with doxycycline or ethidium bromide would be beneficial. Additionally, demonstrating that PQ induces oxidative stress in aged animals would be more convincing that there is a decrease in UPRmt function in these cells.

We thank the reviewer for this important and constructive suggestion. In the responses below, we indicate the changes made to the revised manuscript and Figures.

We agree that validating our findings with an additional UPR^MT^ inducer and directly demonstrating oxidative stress in aged flies enhances the robustness of our conclusions. To address this point, we performed the following additional experiments, now included in Figure 1A-C:

1. Validation of UPR^MT^ induction with doxycycline: We exposed flies expressing the Hsp60::dsRed reporter to doxycycline, a well-established UPR^MT^ inducer. Similar to paraquat (PQ), doxycycline significantly increased reporter activation in the antennal lobe, confirming that the Hsp60::dsRed signal is not specific to PQ but reflects general mitochondrial stress responsiveness.

2. Oxidative stress in aged neurons measured with MitoTimer: We used the GH146-Gal4;UAS-MitoTimer line to directly quantify mitochondrial oxidation in olfactory projection neurons (OPNs). Aged flies displayed significantly elevated MitoTimer red/green ratios, indicating increased mitochondrial oxidative stress in these neurons. This supports our interpretation that reduced Hsp60::dsRed activation in aged animals reflects impaired UPR^MT^ function despite elevated oxidative stress.

3. Functional validation using a second mitochondrial reporter: To further validate the mitochondrial stress response, we employed a second independent reporter, Hsc70-5::dsRed, generated in this study. Like Hsp60::dsRed, this reporter responds to PQ and doxycycline, and its activation is also restored in aged flies upon knockdown of the H3K9 methyltransferase dSetdb1 in OPNs (Figure 4 —figure supplement 2), confirming its regulation via the same UPR^MT^ pathway.

4. Restoration of mitochondrial redox state by dSetdb1 knockdown: Importantly, we also show that dSetdb1 knockdown in OPNs restores mitochondrial redox balance in aged neurons. Specifically, MitoTimer analysis revealed a return to youthful mitochondrial oxidation levels in aged flies with dSetdb1 knockdown, further supporting the idea that activation of UPR^MT^ through epigenetic modulation has a functional impact on mitochondrial health during aging.

In addition to our experimental data, existing literature supports the notion that PQ induces more severe oxidative stress in aged animals, particularly in neurons. In *Drosophila*, older flies show increased sensitivity to PQ-induced lethality, neurodegeneration, and locomotor deficits (Cassar et al., 2015; Neves et al., 2022). Similar age-dependent susceptibility has been observed in mammals, including enhanced dopaminergic neuron loss and behavioral impairments in aged PQ-exposed mice and rats (Rudyk et al., 2017; Somayajulu-Nitu et al., 2009).

Together, these findings reinforce our conclusion that the reduced Hsp60::dsRed and Hsc70-5::dsRed activation in aged neurons reflects a true decline in UPRMT capacity, not insufficient stress induction, and that epigenetic repression via dSetdb1 is a central mechanism underlying this decline.

References added in the revised manuscript:

1. Cassar et al., 2015. doi.org/10.1093/hmg/ddu430

2. Neves et al., 2022. doi.org/10.1016/j.toxlet.2022.03.010

3. Rudyk et al., 2017. doi.org/10.3389/fnagi.2017.00222

4. Somayajulu-Nitu et al., 2009. doi.org/10.1186/1471-2202-10-88

- Regarding panel Figure 1F, it appears the dataset for Ubl RNAi (PQ) and Crc RNAi (PQ) is duplicated. Please verify if this is an error.

The reviewer is correct in identifying the duplicated datasets for Ubl RNAi (PQ) and Crc RNAi (PQ) in Figure 1F. We apologize for this error, which occurred during the data compilation process. We have thoroughly re-examined our raw data and corrected the duplication issue. The statistical analyses have also been performed on the corrected dataset. We can confirm that this correction does not alter the overall conclusions of our study.

(2) Figure 2: Can Setdb1 inhibition restore the ability of aged animals to respond to PQ? This experiment is crucial to support the authors' model.

The reviewer raises a valid point regarding the importance of demonstrating whether dSetdb1 inhibition can restore the ability of aged animals to respond to PQ-induced mitochondrial stress. While Figure 2 describes how dSetdb1 knockdown can allow UPR^MT^ activation without a mitochondrial inducer in aged animals, we agree that directly testing the response to PQ in dSetdb1 loss-of-function (LOF) flies would further strengthen our model. We have already performed a similar experiment, presented in Figure 4F, where we specifically inhibited dSetdb1 in olfactory projection neurons (OPNs) labeled with GFP. This experiment revealed a significant difference in Hsp60::dsRed signal between 45 days post eclosion (dpe) flies treated with PQ, with dSetdb1 knockdown flies showing an increased reporter signal compared to control flies (Mean: Control PQ 378615 vs. dSetdb1 PQ 1587996; Mean diff: -1209381; 95.00% CI of diff. -1710454 to -708307; Adjusted P Value <0.0001). This result suggests that dSetdb1 inhibition can indeed restore the ability of aged neurons to respond to PQ-induced stress. Consistent with this, a similar experiment using our second independent UPR^MT^ reporter, Hsc70-5::dsRed, also demonstrated a significantly enhanced response to PQ in aged (45 dpe) dSetdb1 knockdown flies compared to controls (Mean: Control PQ 391625.56 vs. dSetdb1 PQ 1214149.13; Mean diff: -1157579; 95.00% CI of diff. -1913845 to -401313; Adjusted P Value = 0.0003; Figure 4 —figure supplement 2).

To further support this conclusion at a functional, organismal level, we have included new data robustly validating the behavioral phenotype using the bona fide dSetdb1 loss-of-function (LOF) allele (Figure 4 —figure supplement 1). This experiment clearly shows that aged (45 dpe) dSetdb1 LOF animals exhibit a remarkable preservation of olfactory function compared to the significant decline seen in age-matched controls. We argue that this robust preservation of a complex neurological function is the direct behavioral outcome of the restored cellular stress-response capacity demonstrated in Figure 4F. We believe that the ability of these aged animals to maintain healthy olfactory behavior strongly implies that their neurons have retained the capacity to effectively respond to and manage mitochondrial stress, as our data suggest. Therefore, we believe the combination of the cellular data in Figure 4F and this new functional data in Figure 4 —figure supplement 1 addresses the reviewer's crucial point

- The description of figures 2G-I is biased. For instance, it states, ' In vAChT neurons, dSetdb1 expression remains constant throughout aging (Figure 2G). However, both Utx and Kdm2 exhibit an age-dependent decrease in expression (Figure 2H and I).'. The authors fail to acknowledge the decrease in dSetdb1 in vGlut neurons. Furthermore, the raw data indicates a significant decrease in dSetdb1 (vAChT/15), with a p-value of 0.0042, which is neither represented in the figure nor discussed.

The reviewer is correct that the original description of Figure 2G-I was biased and did not fully represent the data. The statement that "dSetdb1 expression remains constant throughout aging in vAChT neurons" is inaccurate, as the raw data indicates a significant decrease in dSetdb1 expression in these neurons at 15 dpe (p-value = 0.0042). The revised text acknowledges this decrease and provides a more balanced description of the expression patterns of dSetdb1, Utx, and Kdm2 in different neuronal populations during aging. It also highlights the cell-type-specific differences in the regulation of these epigenetic modifiers (Page 6 of the revised manuscript). Modified Text: “In vAChT neurons, dSetdb1 expression remains constant throughout aging (Figure 2G). However, both Utx and Kdm2 exhibit an age-dependent decrease in expression (Figure 2H and I).”

(3) Figure 3: This figure shows that RNAi for dve, ubl, and crc leads to a premature decrease in olfactory function. Is this due to their short lifespan? Would any intervention that decreases lifespan also cause this? The same question applies to dSetdb1: is it long-lived? If so, this could be an alternative explanation for a better olfaction function during aging.

We appreciate the reviewer's critical point regarding the potential influence of lifespan on olfactory function. We have addressed this for each of the genetic manipulations where lifespan was measured. Regarding the premature olfactory decline, our data show this is not a simple artifact of a shortened lifespan. For utx and kdm2, pan-neuronal knockdown does not significantly alter the survival curve compared to controls. While pan-neuronal knockdown of dve does shorten the lifespan, the median survival is approximately 40 days post-eclosion (dpe) (Author response image 1). Crucially, the olfactory deficit in dve knockdown flies is already fully present at 0 dpe, an age at which the population is vigorous. This timing strongly indicates the phenotype is a direct result of the gene's role in neuronal function, not a secondary consequence of accelerated aging. Furthermore, for all aged experiments, only flies that appeared healthy were selected for behavioral assays. Conversely, regarding the preservation of function in dSetdb1 knockdown flies, we found that pan-neuronal knockdown increases the median survival without extending the maximum lifespan (new Figure 3 —figure supplement 1). This pattern is indicative of an improved healthspan, suggesting that animals remain healthier for longer, which aligns with our observation of preserved olfactory preference index. Our key experiment demonstrates that restricting the knockdown of dSetdb1 exclusively to olfactory projection neurons (OPNs) is sufficient to fully recapitulate the preservation of olfactory function in aging (Figure 4L). By confining the genetic manipulation to this specific neuronal population, we can conclude that the effect on olfaction is a cell-autonomous improvement in neuronal health, rather than a systemic effect on longevity.

- Regarding Figure 3D, the source data does not match the quantification. In the extended data for Figure 3A, the gel in the upper left shows that WT has less H3K9me3 than dSetdb1, but this is not reflected in the graph. Please review the quantification.

We thank the reviewer for their careful and diligent inspection of the data. Upon re-examining the figure, we found that the reviewer was correct to point out a discrepancy; however, the issue was not with the quantification itself but with the presentation. The order of the bars in the graph did not match the order of the lanes in the accompanying Western blot gel image, which created a misleading comparison. We have now corrected this presentation error in the revised Figure 2A. The bar graph has been reordered to precisely match the lane order in the Western blot, confirming that our original quantification was correct. To further enhance clarity and transparency, we have also included the full, uncropped membrane images in the supplementary data. We believe this correction and the additional data fully resolve the reviewer's valid concern.

This data is revised in the Results section of the manuscript, page 7.

(4) Figure 4L: This data strongly supports the authors' model, showing that the beneficial effects of dSetdb1 RNAi in rescuing olfactory function depend on UPRmt. Is this effect also observed with 3-octanol?

We thank the reviewer for this excellent question. While the epistasis experiment in Figure 4L was performed with the attractive odorant 2,3-butanedione, we are confident the conclusion applies to aversive odors as well. Our reasoning is based on two key findings presented in our manuscript. First, the age-related olfactory decline and its rescue by dSetdb1 knockdown is a general phenomenon, affecting responses to both attractive (2,3-butanedione) and aversive (3-octanol) odors (Figure 3B, Figure 4 —figure supplement 1). Second, the UPR^MT^ pathway itself is required for general olfactory function, as knockdown of its key components, dve and crc, impairs olfaction in young animals (Figure 3C, Figure 4L). Since both the age-related phenotype and the rescue mechanism are not odorant-specific, we conclude that the UPR^MT^-dependence of the rescue is a general mechanism that would also be observed with 3-octanol.

(5) Figures 5 and 6: These figures are convincing and experiments were well performed. However, including experiments with utx and K_dm_ would further support the model. Would utx and/or K_dm_ knockdown affects mitochondrial function in young animals?

We thank the reviewer for this insightful question. The reviewer correctly suggests that investigating the demethylases Utx and Kdm2 would further support our model. While we have not performed these specific experiments, the existing literature strongly supports the conclusion that knockdown of these demethylases would indeed impair mitochondrial function, consistent with our model. Our data show that pan-neuronal knockdown of Utx or Kdm2 in young flies leads to a premature decline in olfactory function, mimicking the phenotype of aged controls (Figure 3E). This behavioral deficit is accompanied by an increase in global H3K9me3 levels (Figure 3D).

Seminal work in *C. elegans* has firmly established a causal link between these two phenomena. Specifically, Merkwirth et al. (2016) demonstrated that the worm homologs of Utx and Kdm2 (jmjd-3.1 and jmjd-1.2, respectively) are essential for activating the UPR^MT^. Loss of these demethylases prevents the removal of repressive histone marks, thereby suppressing the UPR^MT^ and compromising mitochondrial homeostasis. Furthermore, a study by Yuan et al. (2020) showed that knockdown of the H3K9 methyltransferase set-6 and the epigenetic reader baz-2 improves mitochondrial function in aged worms.

Given this strong, conserved link between H3K9 demethylase function, UPR^MT^ activation, and mitochondrial health, we can confidently infer that the premature olfactory decline we observe in young Utx/Kdm2 knockdown flies is a direct consequence of compromised mitochondrial function. Therefore, the literature provides a solid foundation for our model, and we have updated the discussion to better integrate these key findings.

- In Figure 6H, the values for dSetdb1 RNAi (0) and Control (45) show four duplicated values (99654, 8854, 134193, 227, 97077, 7355, 109124, 186). Please check if this was a mistake.

The reviewer is correct in pointing out the duplicated values in Figure 6H for dSetdb1 RNAi (0) and Control (45). We apologize for this error, which occurred during data compilation. We have carefully re-examined our raw data and confirmed the presence of the duplicated values. To rectify this, we have corrected Figure 6H and the corresponding raw data file which include the statistical analysis. The corrected figure now accurately reflects the actual experimental results and does not alter the original conclusions (Page 24 of the revised manuscript).

We appreciate the reviewer's diligence in identifying this error which went completely unnoticed to us.

Minor comments:

- The order of the bars is confusing because it does not match the order of the Western Blot (WB) samples. Consider reordering for clarity.

We thank the reviewer for pointing out this inconsistency. To improve clarity, we have reordered the bars in the graph to precisely match the lane order of the Western blot samples in the revised figure.Referees cross-commentingI agree with the concerns raised by the other two reviewers, especially the suggestion that Setdb1 levels should be checked to ensure RNAi efficiency. Additionally, using a loss-of-function allele would strengthen the manuscript.

Regarding the efficiency of the dSetdb1 RNAi, we refer the reviewer to our detailed response to Comment 4, where we provide robust in vivo evidence of the tool's functional efficacy by demonstrating its ability to suppress the age-dependent accumulation of H3K9me3, the direct catalytic product of the dSetdb1 enzyme (Figure 2A and Figure 4B). Furthermore, in full agreement with the reviewer's suggestion, we have now validated our key behavioral findings using the classical loss-of-function allele, dSetdb1 lof (egg²³⁵). These new data, presented in Figure 4 —figure supplement 1, show a remarkable consistency between the phenotype observed with the RNAi lines and the loss-of-function allele, providing strong evidence for the specificity of our findings. We are confident that these additions have significantly strengthened the manuscript.

Reviewer #3 (Significance (Required)):Previous studies, especially in worms, have implicated epigenetic alterations in the age-related decline of UPRmt. This study significantly advances our understanding of the mechanistic aspects of olfaction function decay during aging in Drosophila, linking it to mitochondrial proteostasis and epigenetics. This work will be of interest to researchers in the fields of mitochondrial biology, mitochondrial stress signaling (including UPRmt), and aging.My expertise is in proteostasis in the context of aging.